# Aircraft-based observations of isoprene epoxydiol-derived secondary organic aerosol (IEPOX-SOA) in the tropical upper troposphere over the Amazon region

Christiane Schulz[1], Johannes Schneider[1], Bruna Amorim Holanda[1], Oliver Appel[1], Anja Costa[2], Suzane S. de Sá[3], Volker Dreiling[4], Daniel Fütterer[5], Tina Jurkat-Witschas[5], Thomas Klimach[1], Christoph Knote[6], Martina Krämer[2], Scot T. Martin[3,8], Stephan Mertes[9], Mira L. Pöhlker[1], Daniel Sauer[5], Christiane Voigt[5,7], Adrian Walser[5,6,14], Bernadett Weinzierl[5,14], Helmut Ziereis[5], Martin Zöger[4], Meinrat O. Andreae[1,10], Paulo Artaxo[11], Luiz A. T. Machado[12], Ulrich Pöschl[1], Manfred Wendisch[13], and Stephan Borrmann[1,7]

[1]Particle Chemistry, Biogeochemistry and Multiphase Chemistry Departments, Max Planck Institute for Chemistry, Mainz, Germany
[2]Institut für Energie- und Klimaforschung (IEK7), Forschungszentrum Jülich GmbH, Jülich, Germany
[3]School of Engineering and Applied Sciences, Harvard University, Cambridge, Massachusetts, USA
[4]Flight Experiments, German Aerospace Center (Deutsches Zentrum für Luft- und Raumfahrt), Oberpfaffenhofen, Germany
[5]Institute for Atmospheric Physics, German Aerospace Center (Deutsches Zentrum für Luft- und Raumfahrt), Oberpfaffenhofen, Germany
[6]Meteorological Institute, Ludwig Maximilian University, Munich, Germany
[7]Institute for Physics of the Atmosphere, Johannes Gutenberg University, Mainz, Germany
[8]Department of Earth and Planetary Sciences, Harvard University, Cambridge, Massachusetts, USA
[9]Leibniz Institute for Tropospheric Research, Leipzig, Germany
[10]Scripps Institution of Oceanography, University of California San Diego, La Jolla, California, USA
[11]Instituto de Física, Universidade de São Paulo, São Paulo, Brazil
[12]Instituto Nacional de Pesquisas Espaciais (INPE), Centro de Previsão de Tempo e Estudos Climáticos, São José dos Campos, Brazil
[13]Leipzig Institute for Meteorology, University of Leipzig, Leipzig, Germany
[14]Faculty of Physics, University of Vienna, Vienna, Austria

*Correspondence to:* Johannes Schneider (johannes.schneider@mpic.de)

**Abstract.** During the ACRIDICON-CHUVA field project (September - October 2014; based in Manaus, Brazil) aircraft-based in-situ measurements of aerosol chemical composition were conducted in the tropical troposphere over the Amazon using the High Altitude and Long Range Research Aircraft (HALO), covering altitudes from the boundary layer height up to 14.4 km. The submicron non-refractory aerosol was characterized by flash-vaporization/electron impact-ionization aerosol particle mass spectrometry. The results show that significant secondary organic aerosol (SOA) formation by isoprene oxidation products occurs in the upper troposphere, leading to increased organic aerosol mass concentrations above 10 km altitude. The median organic mass concentrations in the upper troposphere above 10 km range between 1.0 and 2.5 µg m$^{-3}$ (referring to standard temperature and pressure; STP) with interquartile ranges of 0.6 to 3.2 µg m$^{-3}$ (STP), representing 78 % of the total submicron non-refractory aerosol particle mass. The presence of isoprene epoxydiol-derived isoprene secondary organic aerosol (IEPOX-SOA) was confirmed by marker peaks in the mass spectra. We estimate the contribution of IEPOX-SOA to the

total organic aerosol in the upper troposphere to be about 20 %. After isoprene emission from vegetation, oxidation processes occur at low altitudes and/or during transport to higher altitudes, which may lead to the formation of IEPOX (one oxidation product of isoprene). Reactive uptake or condensation of IEPOX on pre-existing particles leads to IEPOX-SOA formation and subsequently increasing organic mass in the upper troposphere. This organic mass increase was accompanied by an increase

of the nitrate mass concentrations, most likely due to $NO_x$ production by lightning. Analysis of the ion ratio of $NO^+$ to $NO_2^+$ indicated that nitrate in the upper troposphere exists mainly in the form of organic nitrate. IEPOX-SOA and organic nitrates are coincident with each other, indicating that IEPOX-SOA forms in the upper troposphere either on acidic nitrate particles forming organic nitrates derived from IEPOX or on already neutralized organic nitrate aerosol particles.

## 1 Introduction

Volatile organic compounds (VOCs) emitted by vegetation can lead to the formation of secondary organic aerosol (SOA) through atmospheric oxidation and further chemical processes (e.g., Zhang et al., 2007; Jimenez et al., 2009; Hallquist et al., 2009; Shrivastava et al., 2017). One important VOC is isoprene ($C_5H_8$, 2-methyl-1,3-butadiene), the most abundant non-methane hydrocarbon with a global emission rate of ~500 $\mathrm{Tg\,y^{-1}}$, with a large contribution coming from the Amazon rain forest (Guenther et al., 2012). Also the conditions for photooxidative reactions are favoured in this tropical region. Isoprene is

a short-lived atmospheric gas, which is oxidized in the atmosphere by reactions with the hydroxyl radical (OH), nitrate radical ($NO_3$), or ozone ($O_3$). OH-initiated oxidation leads to isoprene peroxide radicals (ISOPOO). Depending on the $NO_x$ and $HO_x$ concentrations, ISOPOO can react further through different pathways.

For $HO_2$ dominant conditions, meaning conditions with low amounts (< 1 ppb) of NO (Wennberg, 2013), ISOPOO will mainly react with hydroperoxyl radicals ($HO_2$) to the intermediate oligomer hydroxyhydroperoxides (ISOPOOH). Further oxidation

of ISOPOOH may lead to isoprene epoxydiols (IEPOX), which then can partition into the particle phase by condensation or reactive uptake and, thus, SOA derived from IEPOX (IEPOX-SOA) can be formed (e.g., Claeys et al., 2004; Carlton et al., 2009; Paulot et al., 2009; Surratt et al., 2010; Lin et al., 2012; Budisulistiorini et al., 2013; Worton et al., 2013; St. Clair et al., 2016; Liu et al., 2016b). Laboratory studies demonstrate that around 50 % of isoprene-derived particulate matter is associated with IEPOX production and uptake through the $HO_2$ pathway and in the presence of acidic aerosol particles (Liu et al.,

2015). The existence of aerosol as seed particles seems to be necessary (Surratt et al., 2010; Lin et al., 2012), but also the acidity of aerosol can influence the formation yield of IEPOX-SOA. Laboratory and field studies found a correlation between IEPOX-SOA and sulfate, which is related to the acidity of aerosol (Surratt et al., 2007; Budisulistiorini et al., 2013). Although the IEPOX pathway is considered as the dominant one, further laboratory studies show that also other gas-phase reactions of ISOPOOH with multifunctional hydroperoxides contribute to the formation of isoprene-derived SOA (e.g., Krechmer et al.,

2015; Liu et al., 2016b; Riva et al., 2016).

A different oxidation pathway occurs at increased NO concentrations (> 1 ppb) in NO dominant conditions (Wennberg, 2013). Reaction pathways for ISOPOO change towards reactions with NO instead of $HO_2$, leading to hydroxynitrates and/or hydroxyalkoxy radicals, which decompose further to methacrolein (MACR) and methyl vinyl ketone (MVK). Whereas MACR is a

precursor for isoprene-derived SOA, MVK has almost no SOA contribution (Surratt et al., 2006; Liu et al., 2016a).

In the central Amazon region, both $HO_2$ and NO dominated conditions were observed dependent on the time and location of measurements (Kuhn et al., 2010; Andreae et al., 2015; Martin et al., 2016). Thus, isoprene can undergo different gas-phase reactions depending on several conditions, leading either to IEPOX-SOA or to other types of isoprene-derived SOA.

Several field studies for investigating IEPOX-SOA were conducted in different regions. The extensive study by Hu et al. (2015) summarizes those with a focus on IEPOX-SOA measured by aerosol mass spectrometry. The tracer ion at mass to charge ratio (m/z) 82 ($C_5H_6O$) was identified to be related to IEPOX-SOA in combination with the ion at m/z 53 ($C_4H_5$) (Robinson et al., 2011; Allan et al., 2014; Hu et al., 2015). In order to quantify the amount of IEPOX-SOA measured by aerosol mass spectrometry, the ratio of the signal at m/z 82 to the whole organic signal was introduced and is defined as $f_{82}$ (e.g., Budisulistiorini

et al., 2013; Allan et al., 2014; Hu et al., 2015). Background values of $f_{82}$ were calculated on the basis of different field studies worldwide. In areas with predominant monoterpene emissions, the background value is $3.1 \pm 0.6\,‰$ . Areas with a strong biomass burning and urban influence a background value of $1.7 \pm 0.1\,‰$ is reported, whereas ambient organic aerosol with a strong isoprene influence under low NO conditions shows an increased value of $6.5 \pm 2.2\,‰$ (Hu et al., 2015).

The AMAZE-08 campaign focused on SOA production mechanisms at a pristine continental site in the Amazon basin during

the wet season (Martin et al., 2010). The submicron aerosol particles were found to be dominated by secondary organic material (Chen et al., 2009). From positive matrix factorization (PMF) analysis one factor was associated with the reactive uptake of IEPOX to acidic haze, fog or cloud droplets (Chen et al., 2015). Another field campaign that was conducted in the Amazon basin is the GoAmazon2014/5 field campaign (Martin et al., 2016). One focus of GoAmazon2014/5 is the study of aerosol sources and SOA formation and aging comparing the dry and wet season and the influence of urban pollution (Shilling et al.,

2018). Another focus is the investigation of parameters influencing the pathways for isoprene oxidation. Shifts in the prevailing regime of NO or $HO_2$ pathways for isoprene photooxidation in the central region of Amazonia were studied (Liu et al., 2016b). Also, shifts in the production of IEPOX-SOA with changing concentrations of sulfate and $NO_x$ in the boundary layer of central Amazon region were studied as part of GoAmazon2014/5 (de Sá et al., 2017). Increased NO concentrations suppress IEPOX-SOA production. Despite the enhancing effect of increased sulfate concentrations for IEPOX-SOA production, the NO effect

is dominating (de Sá et al., 2017). During another airborne measurement campaign in the Amazon rain forest (SAMBBA), the highest $f_{82}$ values were around $9\,‰$ measured at the top of the boundary layer with a maximum flight altitude of 5 km (Allan et al., 2014). A recent paper by Mei et al. (2018) compares aircraft measurements from the GoAmazon2014/5 campaign with data collected during the ACRIDICON-CHUVA campaign and with measurements taken at a ground station in the Amazon (T3). The results show good agreement for many measured atmospheric parameters and gives the opportunity to validate the

data quality for several measurements conducted on different sampling platforms (Mei et al., 2018).

Very few measurements at altitudes higher than the boundary layer were reported. From single particle mass spectrometric measurements IEPOX sulfate esters were identified in 80 % and 50 % of the analyzed particles measured at altitudes of 5 and 10 km in the tropical free troposphere, respectively (Froyd et al., 2010). In the boundary layer, IEPOX sulfate ester in aerosol particles did not occur. This is explained by a low abundance of acidic aerosol particles acting as seed particles and with a

relatively short time since the emission of isoprene (Froyd et al., 2010). However, with lofting of isoprene and its derivatives

above the boundary layer, it is suggested that IEPOX can partition more efficiently to acidic aerosol particles. The reactive uptake of IEPOX contributed 1-20 % of the tropospheric aerosol mass in the tropics where continental convection was active (Froyd et al., 2010).

The presence of SOA in the tropical upper troposphere can have different effects. For example, it can be entrained into the trop-
ical transition layer (TTL) from where further slow, radiatively-driven lifting could transport them into the lower stratosphere (Fueglistaler et al., 2009). There, and in the TTL region of enhanced new particle formation near the tropopause, the SOA could become part of the global tropical layer of elevated submicron particle abundances (Borrmann et al., 2010; Weigel et al., 2011). It has also been suggested that aerosol formation in the upper troposphere can provide a source for cloud condensation nuclei for lower altitudes in the Amazon region (Wang et al., 2016; Andreae et al., 2018).

Another interesting aspect is the presence of organic nitrates. Many studies reported on the difficulties to measure organic nitrates quantitatively, but suggested also a possibility to estimate the amount or at least the presence of organic nitrates based on the measured ion ratios of $NO^+$ to $NO_2^+$ using aerosol mass spectrometry (e.g., Bruns et al., 2010; Farmer et al., 2010; Rollins et al., 2010; Fry et al., 2013; Ayres et al., 2015; Kiendler-Scharr et al., 2016; Ng et al., 2017; Schneider et al., 2017). Oxidation of VOCs with the highly reactive nitrate radical $NO_3$ can lead to different nitrogen-containing oxidation products that can partition to the aerosol-phase (e.g., Kroll and Seinfeld, 2008; Shrivastava et al., 2017). The nitrate radical oxidation of VOCs can contribute up to 20 % of the global VOC oxidation and is supposed to increase the aerosol mass significantly (Boyd et al., 2015).

Field measurements have shown that the major aerosol-phase product of monoterpene oxidation with nitrate radical is likely a hydroperoxy nitrate ($C_{10}H_{17}NO_5$) whereas the analogous isoprene oxidation product has a contribution of less than 1 % of
the total organic nitrate and occurs more in gas phase (Ayres et al., 2015). Laboratory studies suggested that isoprene-derived organic nitrates are formed from SOA reactions, but undergo substitution reactions in which nitrate is substituted by sulfate (Darer et al., 2011). Studies from the south eastern US showed, that organic nitrate aerosol particles from monoterpenes are strongly influenced by anthropogenic pollutants and may contribute to 19-34 % of the total organic aerosol content (Xu et al., 2015). Polluted urban regions are often dominated by inorganic nitrates (Farmer et al., 2010). However, in rural forested areas
a dominance of particulate organic nitrates formed from oxidation of monoterpenes was reported (Fry et al., 2013). A recent study from the Amazon showed with measurements at ground level that up to 87 % of the total nitrate can be attributed to organic nitrate (de Sá et al., 2018).

This study presents submicron aerosol chemical composition measurements and focuses on the presence of IEPOX-SOA at different altitudes above the Amazon. The analysis herein uses data from the ACRIDICON-CHUVA campaign, which was con-
ducted in September 2014 (Wendisch et al., 2016). The study gives insights into the photooxidative state of organic aerosols, the presence of IEPOX-SOA, and also the presence of particulate organic nitrates. A comparison of these parameters for different altitudes is presented.

## 2 The ACRIDICON-CHUVA campaign and instrumentation on HALO

The aircraft campaign ACRIDICON-CHUVA aimed at the investigation of convective cloud systems to better understand and to quantify aerosol-cloud-interactions and radiative effects of convective clouds. ACRIDICON is the acronym for "Aerosol, Cloud, Precipitation, and Radiation Interactions and Dynamics of Convective Cloud Systems"; CHUVA stands for "Cloud Processes of the Main Precipitation Systems in Brazil: A Contribution to Cloud Resolving Modeling and to the GPM (Global Precipitation Measurement)". This campaign was performed with HALO (High Altitude and Long Range Research Aircraft), which is operated by the German Aerospace Center (DLR). During September and October 2014 (dry season (Andreae et al., 2015)), 14 flights were conducted in the region around Manaus, Brazil, with a radius of 1300 km above the Amazon rain forest and to the Atlantic coast. Altitudes up to 14.4 km were reached. HALO was equipped with instruments for measuring basic meteorological parameters, atmospheric radiation, trace gases, and aerosol properties, such as size, number, and mass concentration and chemical composition. An overview on the campaign, its objectives and the instrumentation properties and uncertainties can be found in Wendisch et al. (2016), Voigt et al. (2017) and Machado et al. (2017).

### 2.1 In-situ aircraft instrumentation

#### 2.1.1 Basic meteorological data

Basic meteorological data were obtained from the BAsic HALO Measurement And Sensor System (BAHAMAS) at 1 s time resolution. BAHAMAS acquires data from air flow and thermodynamic sensors as well as from the aircraft avionics and a high precision inertial reference system to derive basic meteorological parameters like pressure, temperature and the 3-D wind vector as well as aircraft position and altitude. Water vapour mixing ratio and further derived humidity parameters are measured by SHARC (Sophisticated Hygrometer for Atmospheric ResearCh) based on direct absorption measurement by a tunable diode laser (TDL) system. Typical absolute accuracy of the basic meteorological data is 0.5 K for temperature, 0.3 hPa for pressure, 0.4-0.6 m s$^{-1}$ for wind and (5 % $\pm$ 1 ppmv) for water vapour volume mixing ratio.

#### 2.1.2 Aerosol number concentration measurements

Aerosol particle number concentrations were measured using the Aerosol Measurement System (AMETYST). It includes four butanol-based condensation particle counters (modified model 5.410 by Grimm Aerosol Technik, Ainring, Germany) with flow rates of 0.6 and 0.3 l min$^{-1}$, configured with different nominal lower cutoff diameters at 4 nm and 10 nm (set via the temperature difference between saturator and condenser). AMETYST samples behind the HALO Aerosol Submicrometer Inlet (HASI) mounted on top of the fuselage, which provides near-isokinetic sampling of aerosol particles up to diameters of 2-3 µm. During this campaign, sampling line losses and inlet transmission limited the lower size cutoff to about 20 nm in the upper troposphere. As cloud hydrometeors are found to cause artefacts in the detected number concentrations, cloud passages have been removed from the data set. For details see Wendisch et al. (2016) and Andreae et al. (2018).

For particles in the size range between 90 and 600 nm data from an ultra-high sensitivity aerosol spectrometer (UHSAS-A)

that was installed as an underwing probe were analyzed. The measurement system is based on the detection of scattered light from laser illuminated aerosol particles. For the ACRIDICON-CHUVA flights used here (AC07-AC10 and AC15-AC20), the mentioned size range was divided into 66 logarithmic size bins. Data for the other four flights (AC11-AC14) are recorded in a different size binning and not used here. Cloud passages and intervals with sample flow deviations were removed. The UHSAS-A was calibrated using spherical polysterene latex particles.

### 2.1.3 Submicron particle chemical composition measurements

During the ACRIDICON-CHUVA campaign we operated a Compact Time-of-Flight Aerosol Mass Spectrometer (C-ToF-AMS) (Drewnick et al., 2005; Canagaratna et al., 2007; Schmale et al., 2010) to investigate background aerosol composition around deep convective cloud systems. The C-ToF-AMS was connected to the HASI for sampling aerosol particles. The instrument was connected via a 6.25 mm (1/4-inch) stainless steel tubing. Aerosol particles enter the C-ToF-AMS via a constant pressure inlet controlling the volumetric flow into the instrument. In contrast to the classical approach (Bahreini et al., 2008) this custom-made (at the Max Planck Institute for Chemistry) device consists of two plates with a flexible orifice in between and is connected to a rotor. Squeezing or relaxing the orifice leads to a change of the volumetric flow rate into the instrument such that the pressure in the aerodynamic lens can be maintained constant. Thus, a constant mass flow into the instrument is achieved. Therefore, all data reported here represent conditions during the calibration on ground with a pressure of 995 hPa and temperatures of 300 K. These conditions define our standard temperature and pressure (STP). Further details of the constant pressure inlet are subject of a separate publication (Molleker et al., manuscript in preparation). The C-ToF-AMS was operated with a time resolution of 30 s, which corresponds to roughly 6 km flight path. An overall accuracy of about 30 % has been reported in previous studies (Canagaratna et al., 2007; Middlebrook et al., 2012).

The refractory black carbon (rBC) particles were measured with a Single Particle Soot Photometer (SP2, Droplet Measurement Techniques, Longmont, CO, USA). The instrument uses a laser-induced incandescence technique to quantify the mass of rBC in individual aerosol particles (Stephens et al., 2003; Schwarz et al., 2006). Calibrations of the incandescent signal were conducted before, during and after the campaign using size-selected fullerene soot particles. The scattering signal was calibrated using either spherical polystyrene latex size standards or ammonium sulfate particles of different diameters selected by a differential mobility analyzer.

### 2.1.4 NO and NO$_y$ measurements

Measurements of nitric oxide (NO) and total reactive nitrogen (NO$_y$) were conducted by a dual-channel chemiluminescence detector (CLD-SR, Eco Physics). Ambient air is sampled via a standard HALO trace gas inlet with a Teflon tube. For the NO$_y$ channel, the chemi-luminescence detector is combined with a custom-built gold converter that reduces all oxidized reactive nitrogen species to NO (Ziereis et al., 2000). NO from ambient air reacts with O$_3$ produced from an ozone generator in a chamber resulting in excited NO$_2$. The emitted luminescence signal is detected. The detector channel is equipped with a pre-reaction chamber for determining cross-reactions of other compounds in ambient air reacting with O$_3$. The time resolution is

**Table 1.** Relative ionisation efficiencies (RIEs) used for data analysis. RIE$_{NH4}$ and RIE$_{SO4}$ are determined from calibrations with ammonium nitrate and sulfate before, during and after the campaign. RIE$_{NO3}$ and RIE$_{Org}$ are literature values (Alfarra et al., 2004; Canagaratna et al., 2007).

| RIE$_{NO3}$ | RIE$_{NH4}$ | RIE$_{SO4}$ | RIE$_{Org}$ |
|---|---|---|---|
| 1.1 | 3.77 | 0.89 | 1.4 |

s. The precision and accuracy of the measurements depend on the ambient concentrations, with typical values of 5 % and 7 % (NO) and 10 % and 15 % (NO$_y$), respectively.

## 3   Data analysis

### 3.1   General C-ToF-AMS data approaches

The C-ToF-AMS was calibrated with monodisperse ammonium nitrate and sulfate before, during and after the campaign in order to estimate relative ionisation efficiencies (RIE) for nitrate and sulfate. The resulting calibration values can be found in Tab.1.

A value of 0.5 for the collection efficiency (CE) was used for all flights (Matthew et al., 2008; Middlebrook et al., 2012). The main fraction of the measured aerosol mass consists of organic matter (see Sect. 4.2), which has no clear effect on

the CE. Furthermore, large amounts of nitrate that would lead to a higher CE were not encountered during the campaign (Middlebrook et al., 2012).

Detection limits were derived as three times the standard deviation from the background signal according to Reitz (2011). Here, a time-dependent cubic spline function was used to determine a detection limit for each data point. This function was developed at the Max Planck Institute for Chemistry (MPIC) and is introduced and explained in more detail in a PhD thesis

(Reitz, 2011). For every data point of the background signal a third order polynomial (cubic) function is calculated through the four neighbouring points (two before and two behind) while omitting the actual point. Applying this method, all trends from the background signal are excluded and just the short-term noise remains. A quantity $R$ is introduced in the algorithm and characterizes the statistical spread of the noise. $R$ is defined by the squares of the deviation between the omitted centre point and the cubic function along a moving window. To relate this $R$ to the standard deviation ($\sigma$) a proportionality factor of

$\sqrt{18/35}$ is needed (Reitz, 2011). The exact derivation of the proportionality factor will not be explained here, but qualitatively it accounts for the fact that not only the points are affected by noise but also the cubic function itself. Thus, $R$ is larger than the standard deviation ($\sigma$). This calculation also provides a continuous $\sigma$ from which the detection limit ($DL$) can be derived using Eq. 1.

$$DL = 3 \cdot \sqrt{2} \cdot \sigma \tag{1}$$

Averaged over a single flight, the detection limit is around $0.11\,\mu g\,m^{-3}$ for organics, $0.02\,\mu g\,m^{-3}$ for nitrate, $0.03\,\mu g\,m^{-3}$ for sulfate and $0.09\,\mu g\,m^{-3}$ for ammonium. It should be noted here that ammonium can experience additional uncertainty due to cross-relations with water in the fragmentation table (Allan et al., 2004). A vertical profile of the averaged detection limits for all species can be found in the supplement (see Fig. S1).

## 3.2  Oxidation state of the organic aerosol

The oxidation state of the organic aerosol indicates the degree of photochemical aging. It can be determined using the correlation between the ratio of the signal at m/z 44 (mostly $CO_2^+$) to the total organic signal, $f_{44}$, and the ratio of the signal at m/z 43 (mostly $C_2H_3O^+$) to the total organic signal, $f_{43}$ (Ng et al., 2010). These two ions are important tools to identify the photochemical aging of organic aerosol components in the atmosphere. Organic aerosol can be grouped into oxygenated organic aerosol (OOA) and hydrocarbon-like organic aerosol (HOA), whereby OOA is further classified into low-volatile OOA (LV-OOA) and semi-volatile OOA (SV-OOA) (e.g., Alfarra et al., 2004; Zhang et al., 2005; Canagaratna et al., 2007; Zhang et al., 2007; Jimenez et al., 2009; Ng et al., 2011; Ortega et al., 2016). The difference between these two sub-components is represented by the two ions at m/z 43 and m/z 44. These two ions, or more specifically $f_{43}$ and $f_{44}$, change during photochemical aging of organic aerosol, leading to higher $f_{44}$ and lower $f_{43}$ values for LV-OOA than SV-OOA. With increasing photochemical aging, organic aerosol of different origins become more similar in terms of chemistry (Jimenez et al., 2009). We use the ratio $R_{44/43}$ ($f_{44}$ divided by $f_{43}$) to show the effect of photochemical aging on organic aerosol particles dependent on the altitude.

## 3.3  Tracer for isoprene epoxydiol-derived secondary organic aerosol (IEPOX-SOA)

The tracer ion at m/z 82 ($C_5H_6O^+$) is attributed to methylfuran and was identified to be related to IEPOX-SOA (Robinson et al., 2011). Due to thermal vaporisation and subsequent electron impact ionization in the C-ToF-AMS, isoprene photooxidation products in the aerosol are decomposed and lead to an increased signal at m/z 82 (Robinson et al., 2011; Lin et al., 2012). A strong correlation is found with the ion at m/z 53, which corresponds to $C_4H_5^+$ (Robinson et al., 2011; Lin et al., 2012). Calculating the ratio of the signal at m/z 82 to the whole organic signal leads to the fraction $f_{82}$. This fraction is identified as a tracer just for IEPOX-SOA, not for other isoprene-derived SOA from different reaction pathways (e.g., Robinson et al., 2011; Allan et al., 2014; Hu et al., 2015). A study by Hu et al. (2015) found background values of $f_{C_5H_6O^+}$ (ratio of $C_5H_6O^+$ to the total organic signal) for different regions worldwide. For urban and biomass-burning influenced regions an averaged background value of $1.7 \pm 0.1\,‰$ is found. Areas strongly impacted by monoterpene emissions show higher averaged background values of $3.1 \pm 0.6\,‰$. For ambient organic aerosol particles that are influenced by enhanced isoprene emissions and no extensive NO emissions with presence of $HO_2$, values of $6.5 \pm 2.2\,‰$ have been reported (Hu et al., 2015). This enhanced value indicates a strong IEPOX-SOA influence. All the mentioned values are derived from high resolution ToF-AMS (HR-ToF-AMS) data. For C-ToF-AMS data with unit mass resolution, the tracer $f_{82}$ can be used, although interferences from other ions than $C_5H_6O^+$ at m/z 82 are possible. Empirical parameterizations relating $f_{82}$ and $f_{44}$ for unit mass resolution are given by Hu et al. (2015, Appendix A) that will be used here for the further analysis.

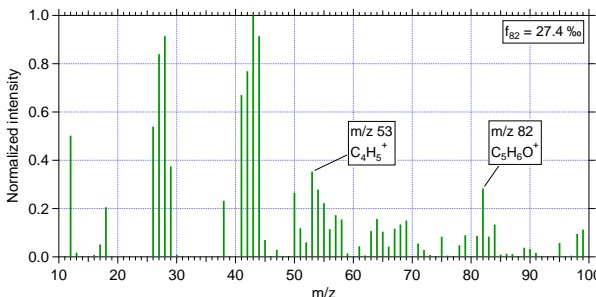

**Figure 1.** Normalized mass spectrum of IEPOX-SOA during flight AC13 conducted on 19 September 2014 at an altitude of 12.6 km. Averaging time for this spectrum was one minute (15:54-15:55 UTC). The two distinct IEPOX-SOA tracers m/z 82 and m/z 53 are labelled. The calculated $f_{82}$ from this mass spectrum is 27.4 ‰.

Figure 1 shows the normalized, measured mass spectrum for a period with high $f_{82}$ during flight AC13 (for the flight notation, see Wendisch et al. (2016)) conducted on 19 September 2014 at an altitude of 12.6 km. The averaging time for this spectrum was one minute (15:54-15:55 UTC). The two peaks for determining IEPOX-SOA, m/z 82 and m/z 53, are labelled and clearly visible in the spectrum. For m/z 82 and m/z 53 the same RIE as for organics is applied (Allan et al., 2014). The calculated $f_{82}$

from this spectrum is 27.4 ‰.

A method for the estimation of the mass concentration of IEPOX-SOA is reported by Hu et al. (2015), see Eq. 2. For this calculation the mass concentration at m/z 82 [$C$(m/z 82)], total organic mass concentration [$C$(Org)], a reference $f_{82}$ value for IEPOX-SOA ($f_{82}^{\mathrm{IEPOXSOA}}$), and a background value ($f_{82}^{\mathrm{Bg}}$) are taken into account. The reference value for $f_{82}^{\mathrm{IEPOXSOA}}$ is set to 22 ‰ (Hu et al., 2015). The background value $f_{82}^{\mathrm{Bg}}$ can be determined with an empirical equation and depends on the influence

of urban, biomass burning or strong monoterpene emissions (Hu et al., 2015). For our data set we assume strong monoterpene emission influence and use Eq. 3 to determine $f_{82}^{\mathrm{Bg}}$ according to Hu et al. (2015, Appendix A).

$$C(\text{IEPOX-SOA}) = \frac{C(\text{m/z 82}) - C(\text{Org}) \cdot f_{82}^{\mathrm{Bg}}}{f_{82}^{\mathrm{IEPOXSOA}} - f_{82}^{\mathrm{Bg}}} \tag{2}$$

$$f_{82}^{\mathrm{Bg}} = 0.0077 - 0.019 \cdot f_{44} \tag{3}$$

In Eq. 2, $C$(m/z 82), and $C$(Org) designate mass concentrations in units of µg m$^{-3}$.

### 3.4   Particulate organic nitrates

Several studies show that qualitative measurements and quantification of organic nitrates are of major interest (e.g., Fry et al., 2009; Bruns et al., 2010; Farmer et al., 2010; Ayres et al., 2015; Kiendler-Scharr et al., 2016; Liu et al., 2017; Schneider et al., 2017). Organic nitrates are decomposed during the evaporation and/or ionisation processes in the C-ToF-AMS and, therefore,

are divided into an organic and a nitrate signal (Farmer et al., 2010). The following use of RONO$_2$ refers to the nitrate content

of organic nitrates, as the organic content cannot be estimated with the described methods. Due to observations during the measurement campaign, the analysis of organic nitrates is described as one part of this study. Four methods are applied to find out whether organic nitrates have been present during the measurements.

– A first estimation of organic nitrates can be derived from the ratio of the nitrate-related ions at m/z 30 ($NO^+$) and m/z 46 ($NO_2^+$). The signal at m/z 30 is mostly from $NO^+$, but also the organic ion $CH_2O^+$ can contribute with a small amount (Allan et al., 2014). Such interferences at m/z 30 with $CH_2O^+$ are corrected in the evaluation software by the fragmentation table (Allan et al., 2004), but it is not possible to distinguish unambiguously between the $NO^+$ and the $CH_2O^+$ ions with a C-ToF-AMS. The signal at m/z 46 is usually dominated by $NO_2^+$ ions (Jimenez et al., 2003; Allan et al., 2004). As organic interferences on the mass spectral signals at m/z 30 (interference from $CH_2O^+$) and m/z 46 (interference from $CH_2O_2^+$) can occur in environments with high biogenic contribution and/or small nitrate concentrations, a correction according to Fry et al. (2018) was applied. The correction of both signals at m/z 30 and 46 is achieved by using correlated organic signals at m/z 29, 42, 43, and/or 45 derived by high resolution measurements. The organic signals at m/z 29 ($CHO^+$) and m/z 45 ($CHO_2^+$) are closest to those affected by the interference and used for the correction here. Equations 4 and 5 give the individual correction for the nitrate signal at m/z 30 and 46, respectively. The correction for $NO^+$ includes the total signal at m/z 30, the default fragmentation correction from the air signal (Allan et al., 2004), and a correction coefficient that depends on the m/z used for the correction ($A_i$). As for m/z 30 the correlated organic signal at m/z 29 is used here, the organic signal at m/z 29 (Org29) needs to be taken into account as well as the contribution of the isotopes of organic CO. For the correction of the nitrate fraction at m/z 46 a term which includes a correlation coefficient $B_i$ and the organic signal at m/z 45 is subtracted from the signal at m/z 46. The correction coefficient $A_i$ is in this case 0.215, $B_i$ is 0.127 (see supplement to Fry et al. (2018)). In the organic signal at m/z 28, 29, 30 and 45 the relative ionization efficiency ($RIE_{Org}$) is already applied and needs to be reversed for the correction of the nitrate signal.

Nitrate fraction at m/z 30:

$$NO^+ = \text{m/z } 30 - 0.0000136 \cdot \text{m/z } 28 - A_i \cdot (\text{Org29} - 0.011 \cdot \text{Org28}) \cdot RIE_{Org} - \text{Org30} \cdot RIE_{Org} \tag{4}$$

Nitrate fraction at m/z 46:

$$NO_2^+ = \text{m/z } 46 - B_i \cdot \text{Org45} \cdot RIE_{Org} \tag{5}$$

The total nitrate signal is then calculated by adding both fractions. The final nitrate mass concentrations were reduced by $0.045\,\mathrm{\mu g\,m^{-3}}$ (STP) on average corresponding to an averaged reduction of 39 % of the initial nitrate mass concentrations. A comparison of the initial and finalized nitrate mass concentrations can be found in the supplement (see Fig. S2).

The ratio of $NO^+$ to $NO_2^+$ is different for inorganic ammonium nitrate and organic nitrate. The ratio for inorganic nitrate

is known from the ionisation efficiency calibration with pure ammonium nitrate. For our instrument its value lies between 1.49 and 1.56 with a mean and standard deviation value of $1.52 \pm 0.03$ and is derived from calibration measurements during the campaign. For organic nitrates the literature presents a range of ratios of $NO^+$ to $NO_2^+$ that are higher than the ratio for inorganic nitrates and lie between 5 and 12.5 (e.g., Farmer et al., 2010; Fry et al., 2009, 2011; Rollins et al., 2009; Bruns et al., 2010; Boyd et al., 2015).

- Second, a range of possible mass concentrations of particulate organic nitrate ($pRONO_2$) can be determined by using C-ToF-AMS data. This range is defined with an upper and lower limit. We calculate the amount of inorganic nitrate from neutralization with ammonium and subtract this value from the measured nitrate. For the upper limit we assume full neutralization of sulfate by ammonium and allow only the remaining excess ammonium to be available for neutralization of nitrate (see Eq. 6). Resulting negative values in the first step (neutralisation of sulfate) mean that not enough ammonium for full neutralisation of sulfate was available and were set to zero. In this case, nitrate could exist as organic nitrate such that the upper limit for organic nitrate equals total measured nitrate. For the lower limit we assumed that nitrate was neutralized by the available amount of ammonium (see Eq.7). Resulting negative values were taken as due to statistical variation. The remaining nitrate is the lowest possible amount of organic nitrate, assuming that particulate nitric acid is not present.

$$\text{Upper Limit: } pRONO_2^{up} = C_{NO3} - \left[ \left( C_{NH4} - \frac{36}{96} \times C_{SO4} \right) \times \frac{62}{18} \right] \tag{6}$$

$$\text{Lower Limit: } pRONO_2^{low} = C_{NO3} - \left[ C_{NH4} \times \frac{62}{18} \right] \tag{7}$$

In Eq. 6 and Eq. 7 $pRONO_2^{up}$, $pRONO_2^{low}$, $C_{NO3}$, $C_{NH4}$, and $C_{SO4}$ designate mass concentrations.

- The third possibility to estimate the mass concentration of the nitrate content of organic nitrates comes from recent studies (Farmer et al., 2010; Kiendler-Scharr et al., 2016) (see Eq. 8). Besides the measured ratio of the mass concentrations for $NO^+$ to $NO_2^+$ ($R_{ambient}$), the ratio of $NO^+$ to $NO_2^+$ derived from calibrations with ammonium nitrate ($R_{cal}$), and a fixed value for the ratio from organic nitrates ($R_{RONO_2}$) are used. In our analyses a value of 10 for $R_{RONO_2}$ was taken as described in Kiendler-Scharr et al. (2016). For a detection limit, Bruns et al. (2010) took $0.1\,\mu g\,m^{-3}$ for a conservative data evaluation of $pRONO_2^{Farmer}$.

$$pRONO_2^{Farmer} = NO_3 \times \left[ \frac{(1 + R_{RONO_2}) \times (R_{ambient} - R_{cal})}{(1 + R_{ambient}) \times (R_{RONO_2} - R_{cal})} \right] \tag{8}$$

- The fourth estimation method is described by Fry et al. (2013) and links the ratios determined for inorganic and organic nitrate. The ratios of $NO^+$ to $NO_2^+$ calculated for inorganic and organic nitrate are instrument specific, but a proportional co-variation is observed. Thus, a 'ratio of the ratios', $\chi$, is proposed and a value for $\chi$ of $2.25 \pm 0.35$ is reported.

$$\chi = \frac{R_{RONO_2}}{R_{cal}} \tag{9}$$

With this $\chi$ and our calibration value $R_{cal}$, a value for $R_{RONO_2}$ is calculated using Eq. 9. The next step puts the derived ratio for organic nitrates into Eq. 8 to determine $pRONO_2^{Fry}$.

It should be emphasized again, that only the mass concentration of the nitrate content of organic nitrate is determined by all of the described methods.

## 4 Results and discussion

Aerosol data obtained during 13 flights of the ACRIDICON-CHUVA campaign were evaluated. Data collected during take-off and landing were removed, and cloud passages were not considered. In the following section, vertical profiles of median and interquartile ranges of different parameters are shown. They are calculated for 500 m altitude bins. Data below 100 m were not considered to eliminate the influence of the airport at take-off and landing. Data above 14 km were not used in the binned vertical profiles due to the low amount of available data points. One flight does not provide any aerosol data (AC10, conducted on 12.09.2014). Therefore, this flight is not included in the analysis of the C-ToF-AMS data. All figures are valid for 13 flights, except where otherwise noted.

### 4.1 Meteorological conditions - boundary layer

The meteorological situation during the ACRIDICON-CHUVA campaign was quite similar for all days. Convection was dominating the daily weather and affecting every flight. The invariance of the meteorological situation is also visible in the temperature profile (see Fig. 2, Panel (**a**)) that barely shows any deviation. An overview of some flight details is provided in Tab. S1. The boundary layer (BL) height was determined with the help of the ambient temperature ($T_{amb}$), the virtual potential temperature ($\Theta_v$), relative humidity with respect to water $RH_w$, and aerosol number concentration for particle diameters larger than 20 nm ($N_{d>20nm}$) (Fig. 2). During the whole campaign the vertical profiles of $T_{amb}$ and $\Theta_v$ showed almost no deviation, indicating stable conditions for the measurement period. The daily evolution of the BL height was noticeable during the flights, which lasted typically seven hours. The maximum heights of the BL varied between 1.2 km and 2.3 km, depending on the flight time. The highest BL heights were measured later in the afternoon. Over the whole measurement period, a mean height of the BL was found to be at $1.7 \pm 0.2$ km (Fig. 2, horizontal dashed line; see also Fig. S3). For the binned vertical profile shown in Fig. 2 the BL height is not so obvious due to smoothing effects while averaging over all flights. The observed BL height is consistent with previous studies in the Amazonian dry season (Andreae et al., 2018). Above the BL is the convective cloud layer, which reached altitudes of about 4-5 km during the campaign (Andreae et al., 2018). The thermal tropopause was at an altitude of 16.9 $\pm 0.6$ km (mean and standard deviation) (Andreae et al., 2018). Therefore, all flights of the ACRIDICON-CHUVA campaign

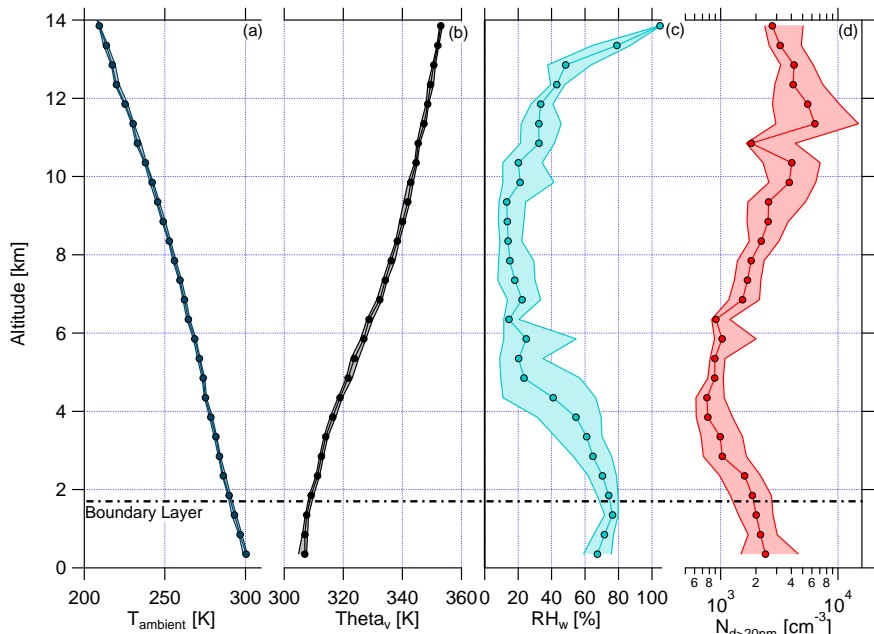

**Figure 2.** Vertical profiles of **(a)** ambient temperature ($T_{amb}$), **(b)** virtual potential temperature ($\Theta_v$), **(c)** relative humidity with respect to water ($RH_w$), and **(d)** aerosol number concentration for particle diameters larger than 20 nm ($N_{d>20nm}$) for all flights during the ACRIDICON-CHUVA campaign. Here medians (connected dots) with interquartile ranges (shaded area) are shown for each plot. The horizontal dashed line shows the mean height of the top of the boundary layer.

were performed in the troposphere.

Figure 2 shows the vertical profiles of median relative humidity with respect to water ($RH_w$) and median aerosol number concentration for particle diameters larger than 20 nm ($N_{d>20nm}$). $RH_w$ decreases above the BL showing a minimum with constant median values between 5 and 9 km. At higher altitudes, $RH_w$ increases again with altitude. The median values of aerosol number concentration are constant in the BL, decrease at middle altitudes with a minimum at 4 km and rise strongly at altitudes above 7 km. This increase was interpreted as evidence for new particle formation at altitudes above 7 km by Andreae et al. (2018).

## 4.2 Aerosol mass concentration

In Fig. 3, the vertical profiles of median mass concentrations of organics, nitrate, sulfate, ammonium, and black carbon given in $\mu g\,m^{-3}$ (STP) are shown, with STP calculated for $T = 300$ K and $p = 995$ hPa from calibration measurements at the ground. At all altitudes, the main fraction of the submicron particulate mass consists of organic matter. The highest aerosol mass concentration, in terms of total aerosol mass as well as median values for all species, are observed at lower altitudes between

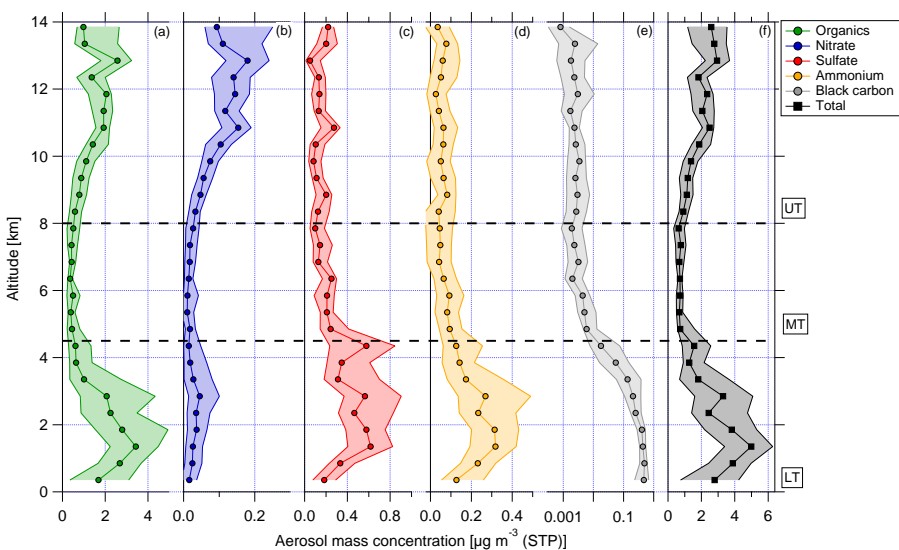

**Figure 3.** Vertical profiles of **(a)** organics (green), **(b)** nitrate (blue), **(c)** sulfate (red), **(d)** ammonium (yellow), **(e)** black carbon (grey), and **(f)** total aerosol (black) median mass concentration and interquartile ranges (in 500 m bins) for 13 flights of the ACRIDICON-CHUVA campaign. Horizontal dashed lines indicate the division into lower, middle and upper troposphere.

0.1 and 4.5 km. This includes the BL, see Sect. 4.1. An exception from that is nitrate, showing maximum median values at altitudes above 10 km. At middle altitudes (between 5 and 8 km) the mass concentrations of all shown species decrease rapidly. A different behaviour of the species is observed at high altitudes between 8 and 14 km. The mass concentrations increase again with increasing altitude for organics and nitrate. For this altitude range, nitrate shows the highest median values. Although
the median values decrease between 13 and 14 km, the 75 percentile range still indicates that high mass concentrations were encountered. In Sect. 4.5 the increase in nitrate mass concentration is discussed with respect to the potential existence of organic nitrates. In contrast to this, the mass concentrations of sulfate and black carbon are highest at lower altitudes, decline above 4.5 km and stay constant at middle and high altitudes. Just between 13 and 14 km, sulfate median values show a slight increase again. At lower altitudes, the mass concentrations of ammonium are highest, above 4.5 km decreasing and staying
constantly low for the rest of the altitude range.

The vertical distribution of aerosol mass concentration allows the classification into three different regions. These are the lower troposphere (LT) ranging from 0.1 to 4.5 km, the middle troposphere (MT) covering altitudes between 4.5 and 8 km and the upper troposphere (UT), which includes altitudes between 8 and 14 km (see Fig. 3).

## 4.3 Oxidation state of the organic aerosol

As described in Sect. 3.2, we calculated the correlation between $f_{43}$ and $f_{44}$. Figure 4, Panel **(a)** presents the data from all flights for the two different altitude regimes LT (0.1 to 4 km) and UT (8 to 14.4 km) averaged over two minutes. The dashed lines represent the region where previous boundary layer aerosol data lie (Ng et al., 2010). The arrow illustrates the direction

(upper left corner) in which data points are 'moving' when photochemical aging occurs. The two different colours indicate the altitude dependency. The dark green markers represent data sampled in the LT, whereas the light green markers show data from the UT. Although there is an overlapping region of both, there is a difference between LT and UT. Most of the data sampled in the LT are located towards the upper left corner of the triangle, meaning that they are more oxidized. In comparison to this,

data from the UT show different properties. Lower $f_{44}$ and at the same time increased $f_{43}$ values show a lower oxidation level of the organic aerosol. Also presented in Fig. 4, Panel **(a)** are the median values with the interquartile ranges for LT and UT, respectively. The median value for the UT has a higher $f_{43}$ and a lower $f_{44}$ than that for the LT. That means organic aerosol measured in the UT is significantly less photooxidized than in the LT. Thus, the organic aerosol particles in LT and UT are different from each other. Some of the LT organic aerosol may be transported to higher altitudes, but most of the organic aerosol

measured in the UT must have a different source that is not located in the BL or in the LT. The possibility that substantial amounts of aerosol are transported from the BL into the UT has been ruled out by the study of Andreae et al. (2018), based on the absence of detectable amounts of black carbon in the UT (also see Fig. 3) and other differences in the properties of aerosol in the LT and UT.

Previous field measurements that have been performed in the Amazon allow a comparison of the presented data set with mea-

surements taken at the ground at two different stations (T3 and T0t). Station T3 is an open field ca. 70 km west of Manaus and has frequent pollution influence from this city, whereas T0t is in a near-pristine rain forest ca. 60 km northwest of Manaus. Thus, they provide anthropogenically influenced or natural measurement conditions, respectively. For further information on the research stations see Martin et al. (2010) and Martin et al. (2016).

The data were collected during the AMAZE-08 and the GoAmazon2014/5 campaigns during the wet season (AMAZE-08:

Feb/Mar 2008; GoAmazon2014/5: Feb/Mar 2014) and represent ground measurements (Chen et al., 2009; de Sá et al., 2018). Figure 4, Panel **(b)** illustrates the median and interquartile ranges for the different data sets. Median values from the GoAmazon2014/5 campaign are similar to the median values derived from the ACRIDICON-CHUVA campaign sampled in the LT, showing that the organic aerosol is oxidized. In comparison, the AMAZE-08 campaign data differ, showing that the organic aerosol during AMAZE-08 was less oxidized. Our data for the UT differ from both these data sets, indicating again a different

source for organic aerosol than in the LT. It should be mentioned here that there is a significant variability of m/z 44 (and $f_{44}$) among different AMS instruments such that no quantitative comparison can be done among the different data sets shown in Fig. 4 (Fröhlich et al., 2015; Crenn et al., 2015; Pieber et al., 2016).

Figure 5 shows the vertical profiles of median and interquartile ranges for $R_{44/43}$, $f_{82}$, calculated IEPOX-SOA mass concentration, and relative contribution of IEPOX-SOA to the organic mass concentration, $R_{\text{IEPOX-SOA/Org}}$, respectively. Panel **(a)**

illustrates the changes of $R_{44/43}$ with altitude. In the BL $R_{44/43}$ is constant, demonstrating that this layer is well mixed. With increasing altitudes this ratio decreases significantly to much lower values than in the LT. The lowest median values are observed in the UT.

This raises the question about the source of the observed organics in the UT. There are three different possibilities. First, horizontal long range transport and a subsequent mixing of these air masses with convectively lofted air could occur. Second,

particles from the near or distant boundary layer might be transported aloft. A third possibility would be in-situ secondary

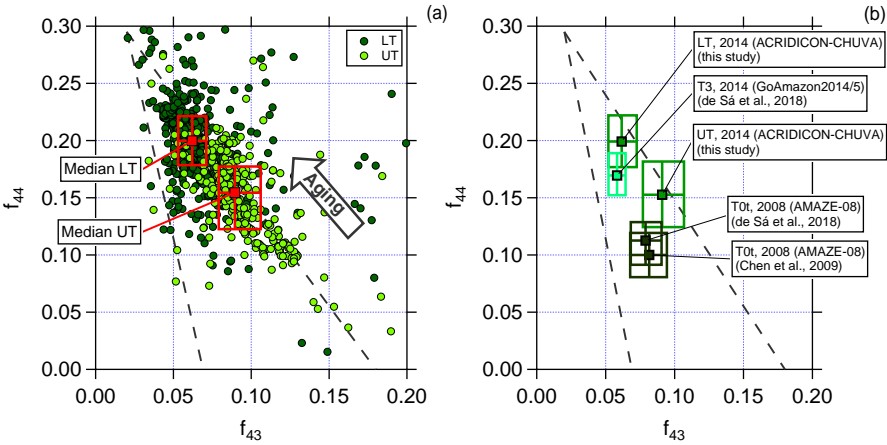

**Figure 4.** Scatter (triangle) plot of $f_{44}$ against $f_{43}$ for **(a)** the lower troposphere (dark green, LT) and upper troposphere (light green, UT) for two minutes averaged data and **(b)** a comparison between different field campaigns performed in the Amazon region. Dashed lines indicate triangular area according to the criteria introduced by Ng et al. (2010). Squared coloured markers and boxes show median values and interquartile ranges for LT and UT, respectively.

organic aerosol formation (SOA) in the UT.

A horizontal long range transport of air masses can be excluded due to the less photochemically aged organics in the UT. The organic aerosol particles in the UT show a lower $R_{44/43}$ ratio meaning that they did not experience much photooxidation, as it would be expected from aerosol influenced by long range transport.

The second possibility, representing the fast convective vertical transport of boundary layer particles, can also be ruled out. The aerosol number concentrations differ considerably (see Fig. 2, Panel **d**) between the LT and UT with strongly increased values in the UT. Furthermore, the sulfate and black carbon aerosol mass concentrations show the highest values in the LT and decrease at higher altitudes (see Fig. 3, Panel **c** and **d**). In case of a fast convective vertical transport of boundary layer particles, the sulfate and black carbon aerosol mass concentrations would show similar values in the LT and UT. The decrease

at altitudes above 4.5 km indicates that the aerosol particles have been efficiently removed (e.g. scavenging) during vertical transport (Andreae et al., 2018).

Air mass trajectories were calculated using the FLEXPART model. The trajectories are calculated along the flight tracks starting every minute and calculated backwards for 10 days providing hourly information on the location of each trajectory. The FLEXPART model is not able to resolve convective transport (see Fig. S4) for the ACRIDICON-CHUVA campaign. Neverthe-

less, the origin of the trajectories that are released in the LT (< 4 km) differs from the origin of the trajectories released in the UT (> 8 km) (see Fig. S5). The trajectories released in the LT have their origin also in the LT and show almost no interaction with higher air masses. Most of the trajectories come from the Atlantic Ocean and the southern part of South America. In contrast to this, the trajectories released above 8 km have their origin mainly above the Pacific Ocean and circulate at high altitudes above South America. Just a minor part of the trajectories origins from the eastern direction, coming from the Atlantic Ocean

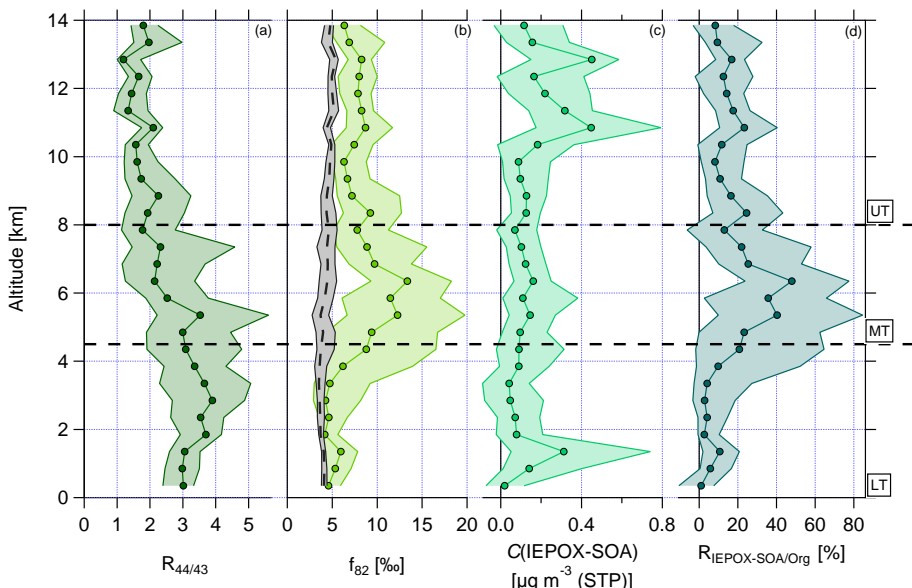

**Figure 5.** Vertical profiles with medians and interquartile ranges of **(a)** $R_{44/43}$, **(b)** $f_{82}$, **(c)** $C(\text{IEPOX-SOA})$, and **(d)** $R_{\text{IEPOX-SOA/Org}}$. Horizontal dashed lines indicate divisions into LT with BL, MT and UT. The vertical dashed line in Panel **(b)** presents the calculated median background values $f_{82}^{Bg}$ with interquartile ranges using Eq. 3. This equation is valid for areas with strong monoterpene influence (Hu et al., 2015, Appendix A).

and/or Africa. Interactions with air masses at lower altitudes are rare, most prominent is the lifting at the Andes mountains. This leads to the conclusion that the third possibility, in-situ SOA formation with subsequent growth of the aerosol particles to large enough sizes that they can be detected by the C-ToF-AMS, is the dominant process in the UT.

Another indication supporting this is the size information of aerosol particles with diameters between 90 and 600 nm. Figure
6 shows the vertical profile of the median and the mode of the binned size distributions measured with the UHSAS-A (Panel **(a)**). It should be noted here, that the lowest cutoff of the considered size range of the UHSAS-A is at 90 nm. Accordingly, the displayed mode diameters are confined by this lower limit. Also, the displayed size distribution medians are affected by the size range limits and should only be interpreted in this context. Whereas in the LT the median and the mode are at diameters around 150 nm (median) and 130 nm (mode), respectively, both the median and the mode are shifted towards smaller diameters with
increasing altitude. The lowest value of the median is reached at altitudes above 4 km and (apparently) remains constant. The colour code in the vertical profile refers to the size distributions for the three different altitude regions in Panel **(b)** of Fig. 6. Shown are the median and interquartile range of the size distributions. The size distribution in the LT shows a maximum at 130 nm. The size distributions in the MT and UT are shifted towards smaller diameters, and it is clearly seen that the highest concentrations of small particles (around 90 nm) are found in the UT.
In the following we present evidence that the formation of IEPOX-SOA in the UT can partly explain this observation.

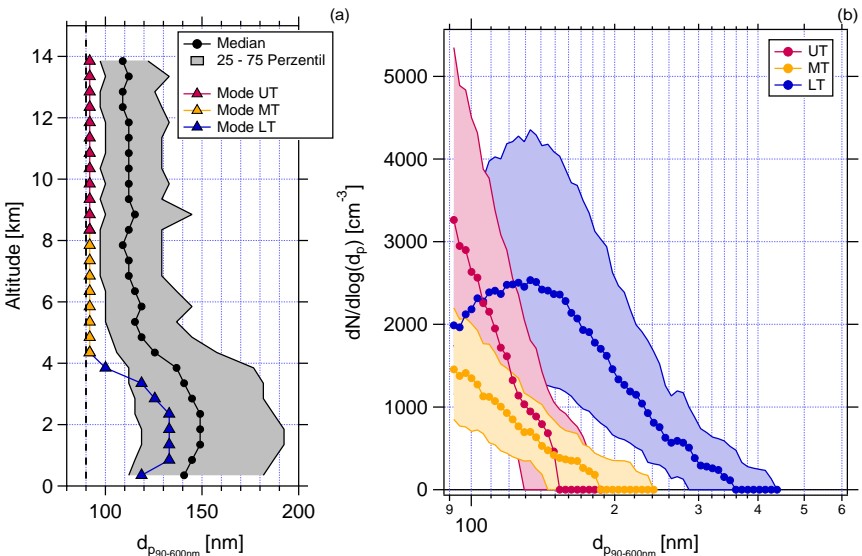

**Figure 6.** Vertical profile of the medians (black dots) and the mode (triangles) of the binned size distributions (Panel **(a)**), and median and interquartile size distributions of particles between 90 and 600 nm in the UT (pink), in the MT (yellow) and in the LT (blue) (Panel **(b)**). The grey area in Panel **(a)** gives the interquartile range. The dotted line in Panel **(a)** indicates the lower cutoff of the considered size range of the UHSAS-A. The statistics shown in both Panels **(a)** and **(b)** are calculated from all valid UHSAS-A data from 10 flights (AC07-AC10, AC15-AC20). Data are calculated for STP conditions.

## 4.4   Observations of isoprene epoxydiol-derived secondary organic aerosol (IEPOX-SOA)

Figure 5 presents the vertical profile of the IEPOX-SOA tracer $f_{82}$ (Panel **(b)**) and the calculated median background values $f_{82}^{Bg}$ (see Eq. 3). The median background values are quite constant between 4 and 5 ‰ over the whole altitude range with small interquartile ranges (see Fig. 5, Panel **(b)**, vertical dashed line with interquartile ranges in grey). This indicates that continuous

emissions and processing of isoprene tend to build an ubiquitous background level up to 14 km. In the LT, $f_{82}$ shows constant median values around 5 ‰, slightly increasing in the interquartile ranges in the upper part of the LT. These values are similar to or higher than the calculated median background values $f_{82}^{Bg}$ suggesting a strong IEPOX-SOA influence (see Sect. 3.3). In the UT, the values of $f_{82}$ are again quite constant over the altitude range, but with increased median values around 8 ‰. The values in the UT lie above the background values and are even higher than in the LT, where an influence of IEPOX-SOA is

observed. This indicates that IEPOX-SOA can have an important impact also in the UT.

Although the characteristics of the MT are not the focus here due to the overall low organic mass concentrations, it should be mentioned that the highest median values of $f_{82}$ were observed in the MT between 4.5 and 8 km. The interquartile ranges are extended here, but the median values reach up to 12 ‰.

To the best of our knowledge, $f_{82}$ data from the tropical upper troposphere have not yet been reported in the literature. Aircraft

measurements above the Amazon rainforest were reported by Allan et al. (2014), but these data are restricted to altitudes

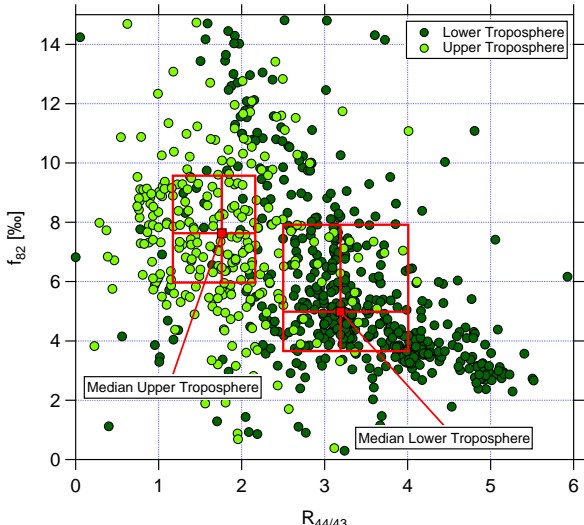

**Figure 7.** Scatter plot of $f_{82}$ against $R_{44/43}$ for the lower troposphere (dark green, LT) and upper troposphere (light green, UT). Data are averaged over two minutes. Red markers and boxes show median values and interquartile ranges for LT and UT, respectively.

below 5 km, corresponding to the definition of the LT in this study. In Allan et al. (2014), two cases are presented and show background (4 ‰) and increased values (9 ‰) of $f_{82}$. The highest values of $f_{82}$ were found on top of the boundary layer, decreasing with increasing altitudes. These values are similar to the data from the LT presented here.

Figure 5, Panel **(c)** depicts the vertical profile of the median IEPOX-SOA mass concentrations, which are calculated using Eq.
2. The lowest values can be found in the LT and MT. However, in the UT, especially at altitudes above 10 km, a strong increase of IEPOX-SOA is observed. Here IEPOX-SOA contributes up to 20 % of the organic mass concentration in the UT (see Fig. 5, Panel **(d)**). The highest contribution of IEPOX-SOA to the organic aerosol mass is observed in the MT, whereas in the LT up to 10 % can be attributed to IEPOX-SOA. This suggests that in the MT as well as in the UT IEPOX-SOA formation can occur. The correlation of $f_{82}$ with $R_{44/43}$ is presented in Fig. 7 using data averaged over two minutes. Again, the two different green
colours refer to the LT (dark green) and the UT (light green). In the LT, high $R_{44/43}$ but low $f_{82}$ values are observed. The $f_{82}$ values are similar to reported ones that describe a strong IEPOX-SOA influence (see Sect. 3.3). Interestingly, the values with low $R_{44/43}$, indicating less photooxidized organic aerosol, are correlated with even higher $f_{82}$.

This suggests that IEPOX-SOA is contributing to in-situ SOA formation in the UT. In general, SOA formation can occur either through new particle formation (NPF) with subsequent growth or by condensation or reactive uptake on pre-existing particles
without NPF. However, at the time that NPF occurs, the aerosol particles would be too small to be measurable with the C-ToF-AMS, implying that growth of these newly formed particles is necessary.

The observed enhanced NO mixing ratios in the UT would likely change the reaction pathway of isoprene to a non-IEPOX route and subsequently no IEPOX-SOA formation would occur. Based on the observed IEPOX-SOA in the UT, the oxidation of isoprene to IEPOX must occur before reaching these high altitudes.

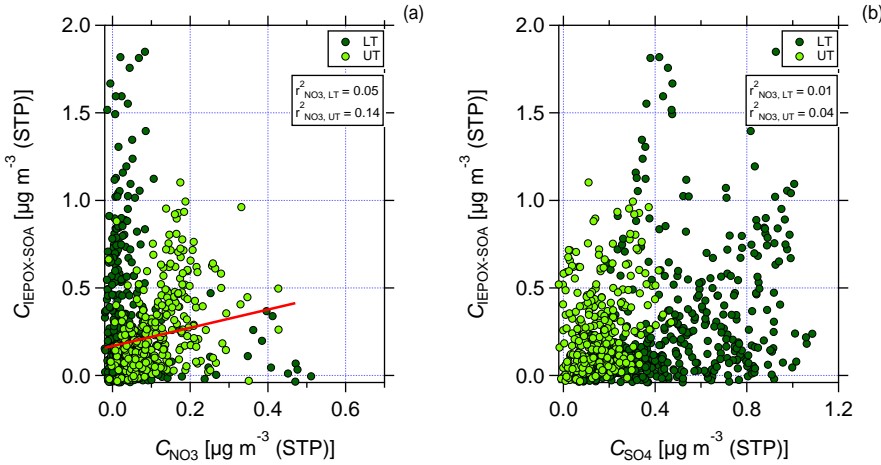

**Figure 8.** Scatter plot of $C$(IEPOX-SOA) against nitrate (Panel **(a)**) and sulfate (Panel **(b)**) mass concentrations for the lower troposphere (dark green, LT) and upper troposphere (light green, UT). Data are averaged over two minutes and presented together with the values for Pearson's $r^2$ for the correlation between $C$(IEPOX-SOA) and nitrate (Panel **(a)**) and sulfate (Panel **(b)**) mass concentration for LT and UT, respectively. In Panel **(a)** linear regression for the correlation between $C$(IEPOX-SOA) and nitrate mass concentration in the UT is presented.

In previous laboratory studies it was found that acidic aerosol is needed for reactive uptake or condensation of gaseous IEPOX onto particles. In the laboratory, the acidic conditions are often realised by using sulfate seed particles (e.g., Surratt et al., 2010; Darer et al., 2011; Lin et al., 2012, 2013). Also in field studies a correlation between IEPOX-SOA and sulfate aerosol was proposed (e.g., Lin et al., 2013; Allan et al., 2014; Hu et al., 2015; Xu et al., 2015; Marais et al., 2016; de Sá et al., 2017).

Figure 8 shows the scatter plot of IEPOX-SOA against nitrate (Panel **(a)**) and sulfate (Panel **(b)**) mass concentrations for LT and UT, respectively. The data are averaged over two minutes. In the LT, no correlation between IEPOX-SOA and nitrate is found. However, in the UT a stronger correlation between IEPOX-SOA and nitrate can be seen (see Fig. 8, Panel **(a)**). For sulfate, the correlation is very low both in the LT and UT (see Fig. 8, Panel **(b)**). This indicates that sulfate might not be necessary for the formation of IEPOX-SOA, but nitrate could be an important, possibly sufficient component in the UT.

Although the correlation between IEPOX-SOA and nitrate is weak, it may indicate that not only sulfate, but also nitrate can provide the acidic conditions for the partitioning of IEPOX to IEPOX-SOA. For our data, taking only the inorganic species (nitrate and sulfate) into account for acidity calculations, the aerosol is mainly neutralized (see Fig. S6). Although there is a tendency in the UT above 10 km that the measured ammonium is not sufficient to neutralize the inorganic species, a quantitative statement cannot be made as the values fall below or close by the DL. The presence of organosulfates and -nitrates could

also affect the acidity calculations. As for the organonitrates the data are already corrected, for organosulfates such a similar correction is not possible with data from a C-ToF-AMS as there are no different fragmentation patterns between inorganic and organic sulfates (Farmer et al., 2010). The partitioning of IEPOX on nitrate aerosol results in organic nitrates. The formation and the estimated amount of organic nitrates are discussed in Sect. 4.5.

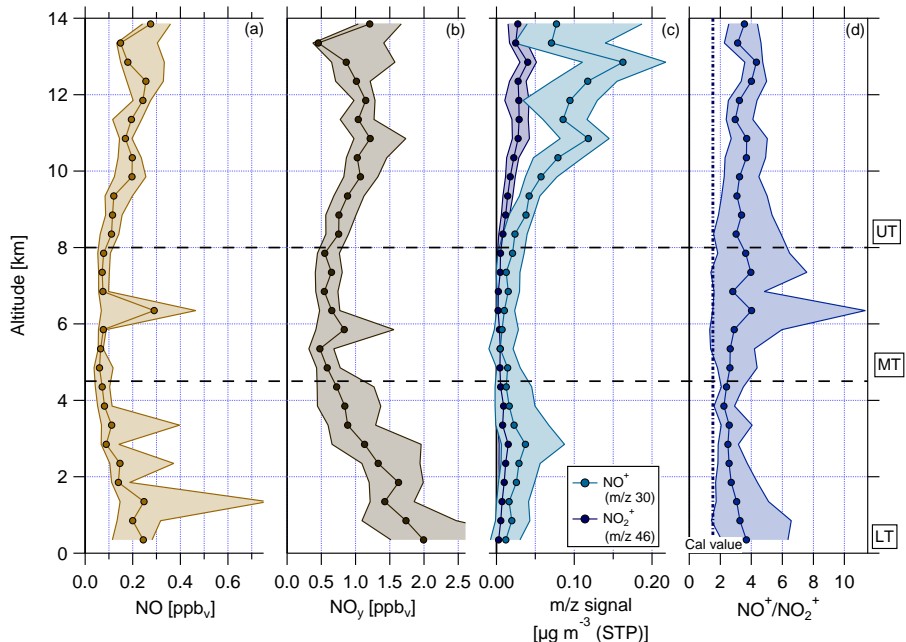

**Figure 9.** Vertical profiles of **(a)** NO mixing ratio, **(b)** reactive nitrogen $NO_y$ mixing ratio, **(c)** $NO^+$ (m/z 30, light blue) and $NO_2^+$ (m/z 46, dark blue) mass signal and **(d)** ratio of $NO^+$ (m/z 30) to $NO_2^+$ (m/z 46). Horizontal dashed lines indicate divisions into LT with BL, MT and UT. The vertical dashed line in Panel **(d)** presents the ratio of $NO^+$ to $NO_2^+$ derived during calibration measurements.

## 4.5 Particulate organic nitrate

The vertical profiles of NO and $NO_y$ mixing ratios are shown in Fig. 9, Panel **(a)** and **(b)**. The mixing ratios of both species are highest in the LT and decrease towards the MT before reaching increased values in the UT again. In the LT, increased $NO_x$ mixing ratios arise from anthropogenic emissions coming from Manaus (Kuhn et al., 2010) and other pollution sources.

A likely source for $NO_x$ in the UT is the production by lightning (Schumann and Huntrieser, 2007). Due to the relatively low $O_3$ values in the UT, $NO_x$ exists mainly in form of NO. After conversion to $NO_y$ (mainly $HNO_3$), $NO_x$ can act here also as a source for (organic) nitrate aerosol and explain the increase in nitrate aerosol mass concentration (see Sect. 4.2).

As already mentioned in Sect. 4.4, the detected ammonium in the presented data is mainly sufficient enough to neutralize the aerosol. There is a tendency that the aerosol is not fully neutralized above 10 km. However, also organics can react with inor-

ganic species forming organic nitrates and sulfates. In the following, the four approaches to estimate the presence of organic nitrates as described in Sect. 3.4 are discussed.

A first estimation of the particulate nitrate content from organic nitrates is the ratio of $NO^+$ to $NO_2^+$. From calibration measurements with ammonium nitrate during the campaign this ratio is known and was found to be in the range between 1.49 and 1.56 with a mean and standard deviation value of $1.52 \pm 0.03$.

Figure 10 shows the scatter plot of the corrected $NO^+$ and $NO_2^+$ for the LT (dark blue) and the UT (light blue). The linear fit

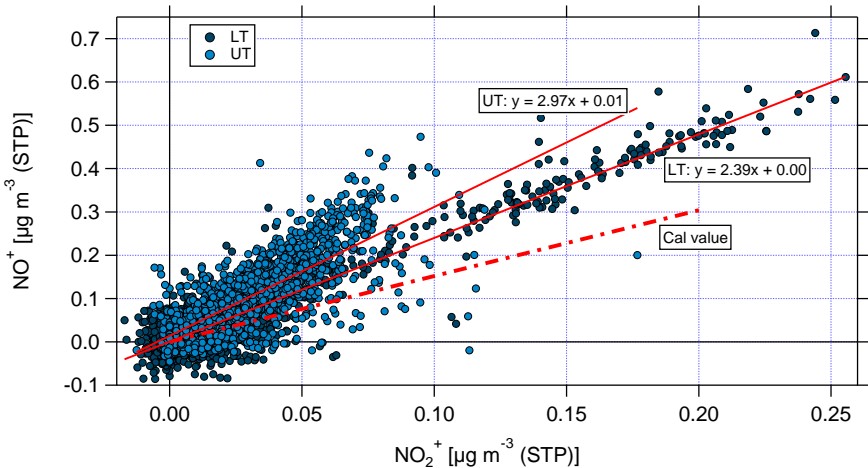

**Figure 10.** Scatter plot of $NO^+$ and $NO_2^+$ for the lower troposphere (LT, dark blue) and the upper troposphere (UT, light blue). The nitrate signals have been corrected for organic interference according to Fry et al. (2018). Linear fit curves are shown in red, the ratio of $NO^+$ and $NO_2^+$ derived from calibrations with ammonium nitrate is presented by the red dashed line.

curves for the LT and UT have an intercept of 0, proving that the applied correction is essential. Nevertheless, it has to be noted that the correction is based on correlations between different m/z signals derived from measurements at low altitudes (Fry et al., 2018), thus the application to UT data bears uncertainties, because the conditions (especially temperature) are different. However, high resolution AMS measurements at these altitudes are currently not available.

The linear fit for the LT data shows a higher slope than that derived from calibrations with ammonium nitrate. The linear fit for the UT shows an even higher slope and some data points are still significantly above the fitted ratio between $NO^+$ and $NO_2^+$. This can be seen as a first hint that organic nitrates might be observed, especially in the UT.

Vertical profiles of median values of m/z 30 and m/z 46 and of the ratio $NO^+$ to $NO_2^+$ are depicted in Fig. 9, Panel **(c)** and **(d)**, respectively. During the whole vertical profile the two lines behave similarly, except for the altitude range between 2 and

6 km, where the distance between them becomes smaller. Compared to the values derived from calibrations with ammonium nitrate, the measured ratios of $NO^+$ to $NO_2^+$ during the flights are much higher and the median values range between 2 and 5. As described in Sect. 3.4, higher $NO^+$ to $NO_2^+$ ratios are linked to organic nitrates and the observed ratio profile can be seen as a first evidence for the presence of organic nitrates.

The second estimation provides a range with a lower and upper limit of nitrate mass concentration of organic nitrates according

to Eq. 6 and 7 (Sect. 3.4). Figure 11, Panel **(a)** shows the estimated lower and upper limits as a vertical profile. In the LT and MT the derived values are below or around zero such that a presence of organic nitrates is unlikely. However, with increasing altitude, the lower and upper limits are also increasing. Especially at altitudes higher than 10 km, both parameters are above zero and the presence of organic nitrates becomes likely. The weakness of this method lies in its dependence on ammonium measurements (see Eq. 6 and 7). Interferences of water in the fragmentation table can lead to a biased estimation of ammonium

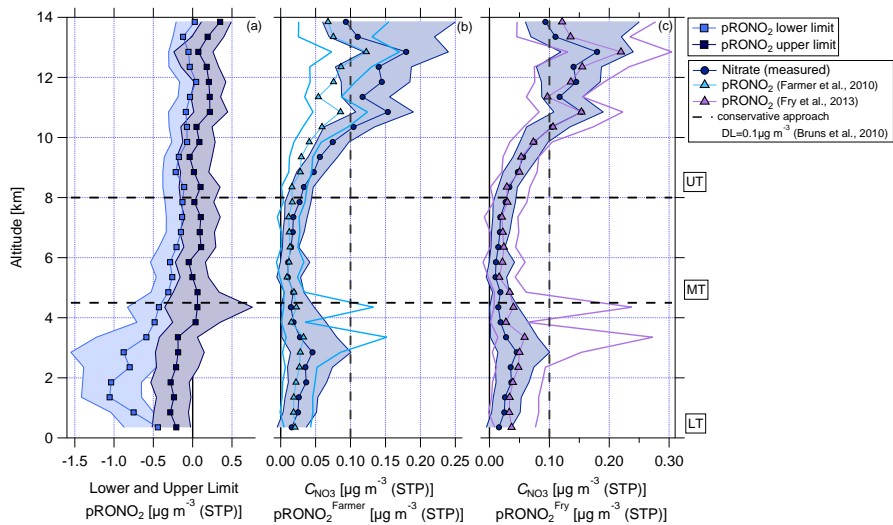

**Figure 11.** Vertical profiles of **(a)** calculated lower and upper limits of organic nitrate mass concentration (approach Sect. 3.4, this study), **(b)** measured nitrate mass concentration and organic nitrate mass concentration calculated according to Farmer et al. (2010), and **(c)** measured nitrate mass concentration and organic nitrate mass concentration calculated according to Fry et al. (2013). Horizontal dashed lines gives divisions into LT with BL, MT and UT. The vertical dashed line in Panel **(b)** shows the conservative detection limit of $0.1\ \mu g\,m^{-3}$ after Bruns et al. (2010).

concentrations (Allan et al., 2004). Biased ammonium concentrations can be one reason for the derived negative values for upper and lower limits of $pRONO_2$.

The third approach uses Eq. 8 to estimate the amount of nitrate mass concentration of organic nitrates. As described in Kiendler-Scharr et al. (2016), a fixed value of 10 for $R_{RONO_2}$ is used here (see Sect. 3.4). In Fig. 11, Panel **(b)** the calculated particulate

nitrate mass concentration of organic nitrates ($pRONO_2^{Farmer}$) is presented. The measured nitrate mass concentration is also shown. The detection limit is set to $0.1\ \mu g\,m^{-3}$, which was reported by Bruns et al. (2010) as a conservative approach. Calculated $pRONO_2^{Farmer}$ are quite similar to the measured nitrate in the vertical profile, although they are slightly smaller, especially in the UT. Values for $pRONO_2^{Farmer}$ in the LT and MT are below the detection limit. Interestingly, at altitudes above 10 km the derived values are partly above the detection limit. This points again to the presence of organic nitrates, especially in the UT at

altitudes above 10 km, but the presence of organic nitrates at altitudes below 10 km cannot be excluded.

The fourth approach is based on Eq. 9 and addresses an estimation of the ratio of organic nitrates from the ratio for inorganic nitrates determined from the usual calibration measurements (see Sect. 3.4). Using our value of $R_{cal} = 1.53 \pm 0.03$ results in $R_{RONO_2} = 3.44 \pm 0.05$ (mean and standard deviation). This value is lower than the ratios reported above from standards and also lower than the ratio that is used for the estimation of Kiendler-Scharr et al. (2016). The vertical profile of $pRONO_2^{Fry}$ is

shown in Fig. 11, Panel **(c)**, again with the measured nitrate mass concentration in the back. The vertical profile of $pRONO_2^{Fry}$ shows similar values than the measured nitrate. At altitudes above 10 km the highest values are reached, suggesting the likely

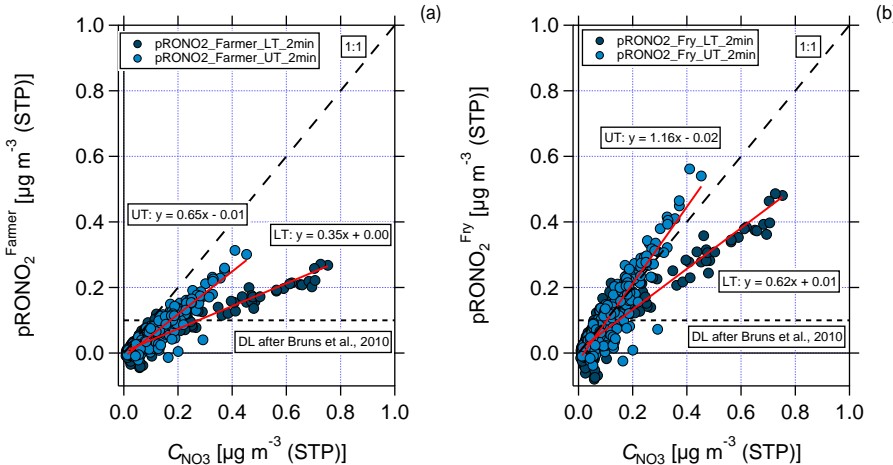

**Figure 12.** Scatter plots of measured nitrate mass concentration against organic nitrate mass concentration calculated according to **(a)** Kiendler-Scharr et al. (2016) and **(b)** Fry et al. (2013) for the lower troposphere (dark blue, LT) and upper troposphere (light blue, UT), respectively. Data are averaged over two minutes. The horizontal dashed line shows the conservative detection limit of $0.1\,\mu g\,m^{-3}$ according to Bruns et al. (2010). The 1:1 line indicates that all particulate nitrate is present as organic nitrate. Also shown are the two fit functions (red) for UT and LT in both plots, respectively.

presence of organic nitrates. Some median values of $\mathrm{pRONO}_2^{\mathrm{Fry}}$ are slightly higher than the measured nitrate concentration, but this lies in the range of uncertainty of the data.

In Fig. 12, a comparison between the measured nitrate mass concentration and the calculated $\mathrm{pRONO}_2$ after Farmer et al. (2010) (Panel **(a)**) and Fry et al. (2013) (Panel **(b)**) is presented. The data are divided into LT (dark blue) and UT (light blue)

and averaged over two minutes. The 1:1 line implies, that all nitrate is present as organic nitrate with a relative fraction of 100 %. The slope of the linear regression that lies below the 1:1 line describes the relative content of nitrate that is present as organic nitrate less than 100 %.

Based on the estimation after Farmer et al. (2010) around 65 % of the measured nitrate data in the UT and 35 % in the LT could be explained with organic nitrates (see Fig. 12, Panel **(a)**). For the comparison with Fry et al. (2013), the whole amount

of measured nitrate in the UT and 62 % of the measured nitrate in the LT could be explained by organic nitrates (see Fig. 12, Panel **(b)**). The slight overestimation for the UT lies in the range of uncertainty of the data. As a qualitative result both comparisons show that in the UT a higher organic nitrate fraction is observed than in the LT.

The quantification of the nitrate content of organic nitrates remains difficult, but a qualitative statement can be made. Methods one, three and four agree with each other that at all altitudes organic nitrates are likely present. In the LT and MT, the calculated

values lie below the detection limit introduced by Bruns et al. (2010). However, in the UT above 10 km the presence of organic nitrates is supported by the results of all four methods.

 Organic sulfates can also be formed in the absence of sufficient ammonium. Farmer et al. (2010) reported that organic sulfates cannot be quantified using C-ToF-AMS data due to a missing tracer ion. Organic sulfates would lead to similar fragmentation

patterns as inorganic sulfates (Farmer et al., 2010). Facing the similar mass concentration of nitrate and sulfate, and especially the increase of nitrate in the UT, the presence of organic nitrates likely plays a similar or even larger role than organic sulfates. The study by Darer et al. (2011) about formation and stability of organic nitrates and sulfates reports that nitrate and sulfates have similar kinetic properties regarding the reaction with tertiary epoxides. According to their study, organic nitrates have shorter lifetimes than organic sulfates and are stable only for short time periods (a few days) before they undergo substitution reactions of nitrate by sulfate or water. Unfortunately, no statement of the temperature dependency of the reactions is given (Darer et al., 2011). The temperatures measured in the UT are low (210-240 K) and could slow down chemical reaction processes and possibly shift the reactions in favour of organic nitrates for a longer time period. In field experiments it was shown that low temperatures shift the chemistry to the formation of organic nitrates (Lee et al., 2014). The condensation of organic nitrates may be important for the growth of newly formed particles in the atmosphere (Berkemeier et al., 2016). Also the phase state of the particles may be important for condensational growth. Over the Amazon basin, SOA particles are predicted to be solid at altitudes above 5 km (Shiraiwa et al., 2017).

## 5 Summary and Conclusion

We presented results from airborne aerosol measurements with a C-ToF-AMS conducted during the ACRIDICON-CHUVA campaign in September and October 2014 in the tropical lower, middle and upper troposphere over the Amazon region. Vertical profiles of the aerosol mass concentrations for organics, nitrate, sulfate, ammonium and black carbon show an overall decrease of the mass concentrations above the lower troposphere. For organics and nitrate the mass concentrations increase again with increasing altitude in the upper troposphere. The characteristics of the organic aerosol were analysed. The photooxidation state of the organics shows a well-mixed lower troposphere with mainly oxidized organics, whereas in the upper troposphere less oxidized organics were observed. Fast vertical transport from the boundary layer or horizontal long-range transport can be excluded as an explanation for this feature. Thus, SOA formation in the upper troposphere is proposed as the most likely process explaining the less oxidized organics. Furthermore, IEPOX-SOA was identified at all altitudes indicating a strong influence on the organic aerosol composition. Previous measurements that reported the enhanced IEPOX-SOA influence in the boundary layer were confirmed (e.g., Allan et al., 2014; Hu et al., 2015; de Sá et al., 2017). In the upper troposphere, the IEPOX-SOA mass concentration increases and is associated with less oxidized organics. This suggests that after emission of isoprene by vegetation, oxidation of isoprene by $HO_2$ proceeds at low altitudes and/or during vertical transport to higher altitudes. The oxidation product, IEPOX, must have been formed before reaching the upper troposphere, where different conditions for the oxidation pathway of isoprene are observed (increased NO and $NO_y$ mixing ratios). In the upper troposphere IEPOX can then partition on pre-existing aerosol particles and thus IEPOX-SOA is formed.

Furthermore, the increase of IEPOX-SOA in the upper troposphere is most likely associated with organic nitrate formation. An increase of nitrate mass concentration was observed in the upper troposphere. Four different methods to estimate the presence and the nitrate mass of organic nitrates were applied and the results support the fact that organic nitrates are present at altitudes above 10 km.

These findings suggest that the formation of IEPOX-SOA and organic nitrates are combined with each other. Two processes could explain the partitioning of IEPOX to the aerosol-phase. The first possibility is that nitrate provides the required acidic conditions and IEPOX is uptaken into the pre-existing aerosol containing organic nitrates. The second possibility is that IEPOX partitions on already neutralized organic nitrates. In this case, the pre-existing aerosol would not need to be acidic.

The formation of IEPOX-SOA in the upper troposphere is an important source for organic aerosol particles at these high altitudes, contributing about 20 % to the total organic aerosol. Further vertical transport of the aerosol particles may lead to entrainment of upper tropospheric aerosol into the tropical tropopause layer (TTL), the "gate to the stratosphere" (Fueglistaler et al., 2009). By this process, organic aerosol particles of tropospheric origin could enter the stratosphere. Also the downward transport of these particles could lead to a regular influence on the aerosol composition at lower altitudes and especially in the boundary layer (Wang et al., 2016; Andreae et al., 2018). This would have effects on cloud condensation nuclei production and thus on cloud properties, and also on the radiative budget.

Emissions from the rainforest influence the tropical atmosphere in several ways and up to high altitudes. The processes described in this study may provide further understanding of the mechanisms that occurred in the pre-industrial tropical atmosphere.

## 6 Data availability

The measured mass concentration data collected with the C-ToF-AMS are available on the HALO data base (HALO-DB). The link is https://halo-db.pa.op.dlr.de/.

*Author contributions.* MOA, PA, LAM, UP and MW designed the research project. VD and MZ provided BAHAMAS data. BW, DS, DF, and AW performed aerosol number concentration measurements and provided AMETYST and UHSAS-A data. BAH and MP provided black carbon data. HZ provided NO and $NO_y$ data. AC, MK, TJW and CV provided cloud data. CS and JS conducted the C-ToF-AMS measurements during the campaign. CS analysed the data with the help of JS and wrote the manuscript. All co-authors commented on the manuscript.

*Acknowledgements.* Special thanks to the whole ACRIDICON-CHUVA team for the successful campaign and fruitful data meetings and discussions. Logistics were handled by DLR-FX; a special thanks for the great support and organisation before, during and after the campaign. Also a special thank to the pilots for realisation of the specific flight patterns. This work was supported by BMBF, grant No. 01LG1205E (ROMIC-SPITFIRE) and by DFG (HALO-SPP 1294, SCHN1138/1-2). Significant funding for the instrument integration, certification, and operation was provided by the support of the Max Planck Society for HALO, as well as from internal resources of the Particle Chemistry Department at Max Planck Institute for Chemistry. P. Artaxo acknowledges FAPESP (Fundação de Amparo à Pesquisa do Estado de São Paulo) grants 2013/05014-0 and FAPESP 2017/17047-0. We thank the LBA program runned by INPA (Brazilian National Institute for Amazonian Research). We also thank CNPq (Conselho Nacional de Desenvolvimento Científico e Tecnológico) for the Expedition Licence 00254/2013-9. We thank Maximilian Dollner and Antonio Spanu for the support during the field measurements. B. Weinzierl, A. Walser, and

D. Sauer have received funding from the Helmholtz Association under Grant VH-NG-606 (Helmholtz-Hochschul-Nachwuchsforschergruppe AerCARE) and from the European Research Council under the European Community's Horizon 2020 research and innovation framework program/ERC Grant Agreement 640458 (A-LIFE). We thank Maximilian Dollner and Antonio Spanu for their assistance during the field measurements. For technical support in the laboratory thanks to Thomas Böttger and Florian Rubach. Also to Franziska Köllner a special thanks for the helpful discussions.

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
