# Peer review of "Aircraft-based observations of isoprene epoxydiol-derived secondary organic aerosol (IEPOX-SOA) in the tropical upper troposphere over the Amazon region"

_Atmospheric Chemistry and Physics, 2018_

## Short Comment (SC1) · 16 May 2018

**Short comment on "Aircraft-based observations of isoprene epoxydiol-derived secondary organic aerosol (IEPOX-SOA) in the tropical upper troposphere over the Amazon region" by Schulz et al.**

(https://doi.org/10.5194/acp-2018-232)

On page 9, lines 11-16, the authors state:
"A first estimation of organic nitrates can be derived from the ratio of the nitrate-related ions at m/z 30 ($NO^+$) and m/z 46 ($NO_2^+$).The signal at m/z 30 is mostly from $NO^+$, but also the organic ion $CH_2O^+$ can contribute with a small amount (Allan et al., 2014). Such interferences at m/z 30 with $CH_2O^+$ are corrected in the evaluation software by the fragmentation table (Allan et al., 2004), but it is not possible to distinguish unambiguously between the $NO^+$ and the $CH_2O^+$ ions with a C-ToF-AMS. The signal at *m/z* 46 is usually dominated by $NO_2^+$ ions (Jimenez et al., 2003; Allan et al., 2004)."

Then the authors proceed to use the default AMS unit mass resolution fragmentation table as an estimate of the $NO^+$ and $NO_2^+$ ion abundances and their ratios to separate ammonium nitrate from organic nitrates. The default unit mass resolution (UMR) fragmentation table assumes only a small interference at *m/z* 30 from $CH_2O^+$ ions (2.2% of OA at *m/z* 29); however, for biogenic SOA the interference can be much larger (of comparable magnitude to OA *m/z* 29). For example, see high-resolution (HR) spectra for biogenic precursor laboratory-generated SOA or ambient PMF OOA factors at the AMS spectral database and references thererin (http://cires1.colorado.edu/jimenez-group/HRAMSsd/) or discussions in the supplementary information of Fry et al. (ACPD 2018). Importantly, for strongly biogenically-influenced rural/remote sites where nitrate is small compared to OA (as in this study), the $CH_2O^+$ ions at *m/z* 30 as well as the $CH_2O_2^+$ ions at *m/z* 46 are comparable or larger than the corresponding nitrate ions, and can lead to large errors in $NO^+$, $NO_2^+$, the $NO_2^+/NO$ ratio, and total nitrate concentrations if not carefully taken into account. To correct for these interferences in UMR data (also with a CToF-AMS) for measurements in the SE United States during summertime (strongly biogenically influenced), an analysis using HR data from a similar region/period was developed, applied and described in detail in Fry et al. (ACPD 2018). Corrections to $NO^+$ and $NO_2^+$ were on average -55% and -33%, respectively, and the reconstructed UMR-based $NO_x+$ ion signals matched the HR signals very well. Thus, it is strongly recommended that the authors either conduct a similar analysis with available HR data collected in the Amazon (e.g., de Sá et al., ACPD, 2018) or the correction from Fry et al. (ACPD 2018) be applied. This correction will likely have large effects on both the inorganic/organic nitrate apportionment and the concentrations (of either type of nitrate and total nitrate).

On a separate but related topic, the paper states that the particles have insufficient amount of ammonium to neutralize the anions present. However, errors in $pRONO_2$ estimation may affect this. In addition, organosulfates (such as from IEPOX) may also be present which may affect this balance. These possibilities should be discussed.

Fry, J. L., Brown, S. S., Middlebrook, A. M., Edwards, P. M., Campuzano-Jost, P., Day, D. A., Jimenez, J. L., Allen, H. M., Ryerson, T. B., Pollack, I., Graus, M., Warneke, C., de Gouw, J. A., Brock, C. A., Gilman, J., Lerner, B. M., Dubé, W. P., Liao, J. and Welti, A.: is a manuscript under review for the journal Atmospheric Chemistry and Physics (ACP). Secondary Organic Aerosol (SOA) yields from NO3

radical + isoprene based on nighttime aircraft power plant plume transects, Atmos. Chem. Phys. Discuss., 1–36, doi:10.5194/acp-2018-255, 2018.

de Sá, S. S., Palm, B. B., Campuzano-Jost, P., Day, D. A., Hu, W., Isaacman-VanWertz, G., Yee, L. D., Brito, J., Carbone, S., Ribeiro, I. O., Cirino, G. G., Liu, Y. J., Thalman, R., Sedlacek, A., Funk, A., Schumacher, C., Shilling, J. E., Schneider, J., Artaxo, P., Goldstein, A. H., Souza, R. A. F., Wang, J., McKinney, K. A., Barbosa, H., Alexander, M. L., Jimenez, J. L. and Martin, S. T.: Urban influence on the concentration and composition of submicron particulate matter in central Amazonia, Atmos. Chem. Phys. Discuss., 1–56, doi:10.5194/acp-2018-172, 2018.

Sincerely,
Doug Day

---

## Referee Comment (RC1) · Anonymous Referee #1 · 17 May 2018

This manuscript presents field results from airborne measurements made over the Amazon in 2014 as part of the ACRIDICON-CHUVA project. It illustrates the potential of using aerosol mass spectrometry instruments to identify ambient SOA formed from IEPOX. Airborne in-situ measurements of aerosol composition and physical properties were used to illustrate the presence of IEPOX-SOA at altitudes > 5 km. Several different approaches are used to quantify the presence of this IEPOX-SOA and subsequently for organic nitrate providing a robust analysis.These observations are original, showing the relationship between organic nitrates and IEPOX-SOA. This manuscript is very well

written, all figures and tables are clear and easily interpreted. This paper is of interest to the ACP audience and is suitable for publication. I have a small number of comments below that can be considered or discussed prior to publication.

Although details of the different flights are provided in other papers (Andreae et al.,) some flight details would be appreciated in the supplementary of this manuscript. This study includes measurements from 13 different flights, how did the meteorological conditions change during each of these flights. According to the overview paper by Andreae et al., 2018, there is some variability linked to air mass source and wind conditions. Can the vertical profiles be classified into different groups depending on meteorological sources?

In section 4.3, the authors state that the sources of the organic aerosol in the LT and the UT are not the same, providing air mass trajectories along the flight track would help support these conclusions.

The authors mention that there are 4 CPC instruments operating during these flights, with cut of diameters of 4 and 10 nm. Were two of the CPCs at 4nm and the other two at 10 nm. In the manuscript Andreae et al.,(2018) it was mentioned that the second set of CPC instruments were coupled with a DMA set up for particle size distribution measurements. Is this also the case for these flights. According to the accompanying papers, other aerosol physical parameters should be available from the UHSAS for larger diameters. These size distribution measurements may help to support the conclusions on the SOA particle growth (Page 15, Line 13).

Page 14, Line 6: It should be mentioned here that there is significant variability of the m/z 44 (or f44) among different AMS instruments and care should be taken when comparing results from different instruments (Frohlich et al., 2015, Pieber et al., 2016, Crenn et al., 2016).

The authors detail several different methods to provide a robust characterization of the presence of organic nitrate and IEPOX SOA during these flights. It could be stated why

[Figure]

PMF analysis was not used to try identify the presence of these aerosols. If IEPOX-SOA is contributing up to 40% of the organic mass, they should be easily extracted by a PMF analysis.

In addition, would adding inorganic ions (SO4 and NO3) into the PMF matrix help in extracting a organic nitrate factor that could then be compared with the other methods used to identify these species?

For the calculation of the organic nitrate concentrations. It is not clear the difference the first estimation and the third estimation. These both methods are based on the ratio of the $NO_+/NO2_+$ ions in the instrument and how it varies from calibration values. However, using the method outlined in Kiendler-Scharr et al., is a more robust and tested method than just using the ratios alone. Can the authors comment on the added values of the first estimation compared to the third?

Clarification on the contents of Figures:

Figure 2: presents data from all flgihts during the ACRIDICON-CHUVA campaign.

Figure 3: presents data from 13 flights of the ACRIDICON-CHUVA campaign.

Figure 4 to Figure 10: It is not stated which flights these measurements correspond.

Can the authors provide more information on which flights were represented in figures 3 to 10 and why they were chosen over all 20 flights?.

---

## Referee Comment (RC2) · Anonymous Referee #2 · 21 Jun 2018

This paper reported an aircraft measurement in the tropical troposphere over the amazon region. The vertical profile of the main components in submicron aerosols were measured by a C-ToF-AMS. The authors focus their discussion on SOA formed through isoprene oxidation under low NO condition, i.e. isoprene epoxydiols-derived SOA (IEPOX-SOA) and organic nitrate formation. Vertical profiles of IEPOX-SOA and organic nitrate mass up to 14 km are new and very interesting. However, the analysis method of this paper has a very serious caveat which needs to be furtherly explored and addressed. I recommend a major revision due to the comments as below:

1) The entire analysis of this paper only depends on AMS measurement. To prove the deduction that the IEPOX-SOA can be formed in the upper troposphere. The authors should show evidences. E.g., what are the vertical profile of isoprene and NO, or even gas-phase IEPOX. The authors argued there is high organic nitrate formation due to NOx production of lighting. However, the high NO will inhibit IEPOX-SOA formation (Paulot, Crounse et al. 2009, de Sá, Palm et al. 2016, Liu, Brito et al. 2016). The addressed reason for organic nitrate formation conflicts with that for IEPOX-SOA formation. And the authors should show recalculate the PH after correcting the mass concentration of nitrate. See detailed information in the followed comment.

2) This paper reported the mass and ions measured with C-ToF-AMS, which gives an UMR spectrum. M/z 30 is mainly composed of $CH_2O^+$ and $NO^+$ ions, and m/z 46 is $NO_2$ and $CH_2O_2^+$. The contribution of $CH_2O$ ions to m/z 30 ($CH_2O_2$ to m/z 46) is very high in areas strongly influenced by biogenic emissions. In the SE US, the aerosol composition of which is very similar to the Amazon area, UMR nitrate (with $NO^+$, $NO_2$, $CH_2O+CH_2O_2$) is overestimated a factor of 2-3 than the high-resolution nitrate (true nitrate with $NO^+$ and $NO_2$ ) based on the default fragmentation table, due to underestimation of the organic ($CH_2O$ and $CH_2O_2$) interferences (Hu, Campuzano-Jost et al. 2017) . Thus the nitrate mass concentration reported in this study are combination of organic and nitrates. As a result, the ratio of UMR m/z 30 and UMR 46 cannot be used to calculate organic nitrate mass concentration, unless the authors find a way to exclude the interference. A revised fragmentation table is suggested in (Allan, Morgan et al. 2014). However its suitability for Amazon area still needs to be evaluated. Similarly, the authors cannot really report the IEPOX-SOA increases with nitrate, which is probably organics. The statement in page 17 1-7 lines are wrong due to the reason mentioned here.

3) Page 20 line 15, tthere is no way that the calculated organic nitrate mass concentration based on Fry et al. (2013) can be higher than the total nitrate mass concentration measured from AMS. Because Fry et al. (2013) used a method base on splitting the

mass of total nitrate masses with differential NO/NO2 ratios from NH4NO3 and organic nitrate. Please check furtherly.

4) How are the vertical profiles of main components in submicron aerosols in this study compared to the amazon study reported in (Allan, Morgan et al. 2014). Also the It will be interesting if the authors can address the similarity and differences, especially for m/z 82 and f82. What is the new finding in this study compared to the previous ones.

Minor comments:

Page 7 line 12: please give details about the "a time-dependent cubic spline function was used to determine a detection limit for each data point"

Page 6 line 13: give full citation of Molleker et al. 2008.

Page 8 line 15-16: Hu et al. 2015 showed an f82 values from IEPOX-SOA and background sources in the appendix.

Page 8 equation 1, what is the f82 values of IEPOX-SOA used in this study to calculate the mass concentration of IEPOX-SOA.

References:

Allan, J. D., W. T. Morgan, E. Darbyshire, M. J. Flynn, P. I. Williams, D. E. Oram, P. Artaxo, J. Brito, J. D. Lee and H. Coe (2014). "Airborne observations of IEPOX-derived isoprene SOA in the Amazon during SAMBBA." Atmos. Chem. Phys. **14**(20): 11393-11407.

de Sá, S. S., B. B. Palm, P. Campuzano-Jost, D. A. Day, W. Hu, M. K. Newburn, J. Brito, Y. Liu, G. Isaacman-VanWertz, L. D. Yee, A. H. Goldstein, P. Artaxo, R. Souza, A. Manzi, J. L. Jimenez, M. L. Alexander and S. T. Martin (2016). "Mass spectral observations of fine aerosol particles and production of SOM at an anthropogenically influenced site during GoAmazon2014 wet season." Atmos. Chem. Phys. Discuss.

Hu, W., P. Campuzano-Jost, D. A. Day, P. Croteau, M. R. Canagaratna, J. T. Jayne, D. R. Worsnop and J. L. Jimenez (2017). "Evaluation of the new capture vaporizer for aerosol mass spectrometers (AMS) through field studies of inorganic species." Aerosol Science and Technology **51**(6): 735-754.

Liu, Y., J. Brito, M. R. Dorris, J. C. Rivera-Rios, R. Seco, K. H. Bates, P. Artaxo, S. Duvoisin, F. N. Keutsch, S. Kim, A. H. Goldstein, A. B. Guenther, A. O. Manzi, R. A. F. Souza, S. R. Springston, T. B. Watson, K. A. McKinney and S. T. Martin (2016). "Isoprene photochemistry over the Amazon rainforest." Proceedings of the National Academy of Sciences.

Paulot, F., J. D. Crounse, H. G. Kjaergaard, A. Kürten, J. M. St. Clair, J. H. Seinfeld and P. O. Wennberg (2009). "Unexpected Epoxide Formation in the Gas-Phase Photooxidation of Isoprene." Science **325**(5941): 730-733.

---

## Author Comment (AC1) · 14 Aug 2018

We thank Douglas Day for his short comment that helped to improve the manuscript.
Our response is formatted as follows:

**Short comment**

Author's reply

Changes to the manuscript

All page, line, section and figure numbers in bold refer to the original manuscript, all others to the revised version.

**Short comment on "Aircraft-based observations of isoprene epoxydiol-derived secondary organic aerosol (IEPOX-SOA) in the tropical upper troposphere over the Amazon region" by Schulz et al.** (https://doi.org/10.5194/acp-2018-232)

**On page 9, lines 11-16, the authors state:**
**"A first estimation of organic nitrates can be derived from the ratio of the nitrate-related ions at m/z 30 ($NO^+$) and m/z 46 ($NO_2^+$). The signal at m/z 30 is mostly from $NO^+$, but also the organic ion $CH_2O_+$ can contribute with a small amount (Allan et al., 2014). Such interferences at m/z 30 with $CH_2O_+$ are corrected in the evaluation software by the fragmentation table (Allan et al., 2004), but it is not possible to distinguish unambiguously between the $NO_+$ and the $CH_2O_+$ ions with a C-ToF-AMS. The signal at _m/z_ 46 is usually dominated by $NO_{2+}$ ions (Jimenez et al., 2003; Allan et al., 2004)."**
**Then the authors proceed to use the default AMS unit mass resolution fragmentation table as an estimate of the $NO_+$ and $NO_{2+}$ ion abundances and their ratios to separate ammonium nitrate from organic nitrates. The default unit mass resolution (UMR) fragmentation table assumes only a small interference at _m/z_ 30 from $CH_2O_+$ ions (2.2% of OA at _m/z_ 29); however, for biogenic SOA the interference can be much larger (of comparable magnitude to OA _m/z_ 29). For example, see high-resolution (HR) spectra for biogenic precursor laboratory-generated SOA or ambient PMF OOA factors at the AMS spectral database and references thererin (http://cires1.colorado.edu/jimenez-group/HRAMSsd/) or discussions in the supplementary information of Fry et al. (ACPD 2018). Importantly, for strongly biogenically-influenced rural/remote sites where nitrate is small compared to OA (as in this study), the $CH_2O_+$ ions at _m/z_ 30 as well as the $CH_2O_{2+}$ ions at _m/z_ 46 are comparable or larger than the corresponding nitrate ions, and can lead to large errors in $NO_+$, $NO_{2+}$, the $NO_{2+}$/NO ratio, and total nitrate concentrations if not carefully taken into account. To correct for these interferences in UMR data (also with a CToF-AMS) for measurements in the SE United States during summertime (strongly biogenically influenced), an analysis using HR data from a similar region/period was developed, applied and described in detail in Fry et al. (ACPD 2018). Corrections to $NO_+$ and $NO_{2+}$ were on average -55% and -33%, respectively, and the reconstructed UMR-based $NO_x+$ ion signals matched the HR signals very well. Thus, it is strongly recommended that the authors either conduct a similar analysis with available HR data collected in the Amazon (e.g., de Sá et al., ACPD, 2018) or the correction from Fry et al. (ACPD 2018) be applied. This correction will likely have large effects on both the inorganic/organic nitrate apportionment and the concentrations (of either type of nitrate and total nitrate).**

We corrected our data according to Fry et al., 2018. For the correction of m/z 30 and 46 the organic signal at m/z 29 and 45 were used, respectively. The correction of the nitrate mass concentration resulted in a reduction of about 39 % (on average) of the initial nitrate mass

concentration. However, there is no change of the general shape of the vertical profile, such that the finding of a nitrate increase in the UT remains unchanged.
We added the following to the manuscript in **Sect. 3.4 p. 9, line 16** (p. 10, line 9):

As organic interferences on the mass spectral signals at m/z 30 (interference from $CH_2O^+$) and m/z 46 (interference from $CH_2O_2^+$) can occur in environments with high biogenic contribution and/or small nitrate concentrations, a correction according to Fry et al. (2018) was applied. The correction of both signals at m/z 30 and 46 is achieved by using correlated organic signals at m/z 29, 42, 43, and/or 45 derived by high resolution measurements. The organic signals at m/z 29 ($CHO^+$) and m/z 45 ($CHO_2^+$) are closest to those affected by the interference and used for the correction here. Equations 4 and 5 give the individual correction for the nitrate signal at m/z 30 and 46, respectively. The correction for $NO^+$ includes the total signal at m/z 30, the default fragmentation correction from the air signal (Allan et al., 2004), and a correction coefficient that depends on the m/z used for the correction ($A_i$). As for m/z 30 the correlated organic signal at m/z 29 is used here, the organic signal at m/z 29 (Org29) needs to be taken into account as well as the contribution of the isotopes of organic CO. For the correction of the nitrate fraction at m/z 46 a term which includes a correlation coefficient $B_i$ and the organic signal at m/z 45 is subtracted from the signal at m/z 46. The correction coefficient $A_i$ is in this case 0.215, $B_i$ is 0.127 (see supplement to Fry et al., 2018). In the organic signal at m/z 28, 29, 30 and 45 the relative ionization efficiency ($RIE_{Org}$) is already applied and needs to be reversed for the correction of the nitrate signal.

Nitrate fraction at m/z 30:
$$NO^+ = m/z\ 30 - 0.0000136*m/z\ 28 - A_i * (Org29 - 0.011*Org28) * RIE_{Org} - Org30*RIE_{Org} \quad (4)$$

Nitrate fraction at m/z 46:
$$NO_2^+ = m/z\ 46 - B_i * Org45 * RIE_{Org} \quad (5)$$

The total nitrate signal is then calculated by adding both fractions. The final nitrate mass concentrations were reduced by 0.045 $\mu g\ m^{-3}$ (STP) on average corresponding to an averaged reduction of 39 % of the initial nitrate mass concentrations. A comparison of the initial and finalized nitrate mass concentrations can be found in the Supplement (see Fig. S2).

Furthermore, we added the following to the paragraph in **Sect. 4.5 p. 19, line 2** (p. 21, line 16):

Figure 10 shows the scatter plot of the corrected $NO^+$ and $NO_2^+$ for the LT (dark blue) and the UT (light blue). The linear fit curves for the LT and UT have an intercept of 0, proving that the applied correction is essential. Nevertheless, it has to be noted that the correction is based on correlations between different m/z signals derived from measurements at low altitudes (Fry et al., 2018), thus the application to UT data bears uncertainties, because the conditions (especially temperature) are different. However, high resolution AMS measurements at these altitudes are currently not available.
The linear fit for the LT data shows a higher slope than that derived from calibrations with ammonium nitrate. The linear fit for the UT shows also a higher slope and some data points are significantly above the fitted ratio between $NO^+$ and $NO_2^+$. This can be seen as a first hint that organic nitrates might be observed, especially in the UT.

[Figure]

Figure 10. Scatter plot of $NO^+$ and $NO_2^+$ for the lower troposphere (LT, dark blue) and the upper troposphere (UT, light blue). The nitrate signals have been corrected for organic interference according to Fry et al., 2018. Linear fit curves are shown in red, the ratio of $NO^+$ and $NO_2^+$ derived from calibrations with ammonium nitrate is presented by the red dashed line.

We added the following to the supplement:

[Figure]

Figure S2. Vertical profile of median nitrate mass concentrations and interquartile ranges before (light blue) and after (dark blue) corrections according to Fry et al., 2018 and supplement to Fry et al., 2018.

**On a separate but related topic, the paper states that the particles have insufficient amount of ammonium to neutralize the anions present. However, errors in pRONO₂ estimation may affect this. In addition, organosulfates (such as from IEPOX) may also be present which may affect this balance. These possibilities should be discussed.**

We updated the neutralization calculation and the corresponding figure (**Fig. S3** (Fig. S6)). The aerosol is mainly neutralized although there seems to be a tendency of acidic aerosol at altitudes above 10 km. We changed the corresponding paragraph to the following in **Sect. 4.4 p. 17, line 2** (p. 20, line 8):

Although the correlation between IEPOX-SOA and nitrate is weak, it may indicate that not only sulfate, but also nitrate can provide the acidic conditions for the partitioning of IEPOX to IEPOX-SOA. For our data, taking only the inorganic species (nitrate and sulfate) into account

for acidity calculations, the aerosol is mainly neutralized (see Fig. S6). Although there is a tendency in the UT above 10 km that the measured ammonium is not sufficient to neutralize the inorganic species, a quantitative statement cannot be made as the values fall below or close by the DL. The presence of organosulfates and -nitrates could also affect the acidity calculations. As for the organonitrates the data are already corrected, for organosulfates such a similar correction is not possible with data from a C-ToF-AMS as there are no different fragmentation patterns between inorganic and organic sulfates (Farmer et al., 2010).

**References:**

de Sá, S. S., Palm, B. B., Campuzano-Jost, P., Day, D. A., Hu, W., Isaacman-VanWertz, G., Yee, L. D., Brito, J., Carbone, S., Ribeiro, I. O., Cirino, G. G., Liu, Y. J., Thalman, R., Sedlacek, A., Funk, A., Schumacher, C., Shilling, J. E., Schneider, J., Artaxo, P., Goldstein, A. H., Souza, R. A. F., Wang, J., McKinney, K. A., Barbosa, H., Alexander, M. L., Jimenez, J. L., and Martin, S. T.: Urban influence on the concentration and composition of submicron particulate matter in central Amazonia, Atmos. Chem. Phys. Discuss., https://doi.org/10.5194/acp-2018-172, in review, 2018.

Fry, J. L., Brown, S. S., Middlebrook, A. M., Edwards, P. M., Campuzano-Jost, P., Day, D. A., Jimenez, J. L., Allen, H. M., Ryerson, T. B., Pollack, I., Graus, M., Warneke, C., de Gouw, J. A., Brock, C. A., Gilman, J., Lerner, B. M., Dubé, W. P., Liao, J., and Welti, A.: Secondary Organic Aerosol (SOA) yields from $NO_3$ radical + isoprene based on nighttime aircraft power plant plume transects, Atmos. Chem. Phys. Discuss., https://doi.org/10.5194/acp-2018-255, in review, 2018.

---

## Author Comment (AC2) · 14 Aug 2018

We thank Referee #1 for her/his comments and suggestions that helped to improve the manuscript. Our response is formatted as follows:

**Reviewer's comments**

Author's reply

Changes to the manuscript

All page, line, section and figure numbers in bold refer to the original manuscript, all others to the revised version.

**This manuscript presents field results from airborne measurements made over the Amazon in 2014 as part of the ACRIDICON-CHUVA project. It illustrates the potential of using aerosol mass spectrometry instruments to identify ambient SOA formed from IEPOX. Airborne in-situ measurements of aerosol composition and physical properties were used to illustrate the presence of IEPOX-SOA at altitudes > 5 km. Several different approaches are used to quantify the presence of this IEPOX-SOA and subsequently for organic nitrate providing a robust analysis. These observations are original, showing the relationship between organic nitrates and IEPOX-SOA.**
**This manuscript is very well written, all figures and tables are clear and easily interpreted. This paper is of interest to the ACP audience and is suitable for publication. I have a small number of comments below that can be considered or discussed prior to publication.**

We thank the reviewer for this positive rating of our manuscript.

1. **Although details of the different flights are provided in other papers (Andreae et al., 2018) some flight details would be appreciated in the supplementary of this manuscript. This study includes measurements from 13 different flights, how did the meteorological conditions change during each of these flights. According to the overview paper by Andreae et al., 2018, there is some variability linked to air mass source and wind conditions. Can the vertical profiles be classified into different groups depending on meteorological sources?**

The meteorological situation was quite similar for all campaign days and dominated by convection. We added an overview table to the supplement giving information on the different flights regarding the duration, meteorological situation, and flight strategy. Air mass sources were analyzed with the model FLEXPART and addressed in the reply to question 2 of the reviewer. The following was added to **Sect. 4.1, p. 11, line 8** (p. 12, line 16):

The meteorological situation during the ACRIDICON-CHUVA campaign was quite similar for all days. Convection was dominating the daily weather and affecting every flight. The invariance of the meteorological situation is also visible in the temperature profile (see Fig. 2, Panel **(a)**) that barely shows any deviation. An overview of some flight details is provided in Tab. S1.

We included the following to the Supplement:

Tab. S1: Overview of all flights with date, duration, maximum altitude that was reached and the meteorological situation. Furthermore, information on the flight strategy and comments to the C-ToF-AMS measurements.

| FLIGHT NO. | DATE IN 2014 | DURATION | ALTITUDE [KM] | METEOROLOGICAL CHARACTERISTICS | FLIGHT STRATEGY AND COMMENTS |
|---|---|---|---|---|---|
| AC07 | 06.09. | 7 h 35 min | 13.9 | Convection | Cloud profiling; no zero filter measurements |
| AC08 | 09.09. | 5 h 30 min | 13.8 | Convection | Cloud profiling |
| AC09 | 11.09. | 6 h 10 min | 12.6 | Convection | Cloud profiling; no zero filter measurements |
| AC10 | 12.09. | 7 h 25 min | 14.4 | Convection, cirrus | Cirrus sampling; CVI measurements only |
| AC11 | 16.09. | 7 h 25 min | 12.9 | Convection | In- & outflow measurements, cloud profiling |
| AC12 | 18.09. | 6 h 15 min | 13.8 | Convection | Polluted cloud profiling |
| AC13 | 19.09. | 6 h 30 min | 12.9 | Convection | Polluted cloud profiling |
| AC14 | 21.09. | 7 h 15 min | 15.2 | Convection | No filter measurements |
| AC15 | 23.09. | 7 h 20 min | 13.8 | Convection | Outflow sampling, cloud profiling |
| AC16 | 25.09. | 6 h 50 min | 13.2 | Convection | In- & outflow measurements |
| AC17 | 27.09. | 6 h 40 min | 8.1 | Convection | Cloud contrast measurements: clouds above forested and deforested areas |
| AC18 | 28.09. | 6 h 50 min | 14.4 | Convection | Clean cloud profiling |
| AC19 | 30.09. | 7 h 15 min | 13.8 | Convection, pyro-Cumulus | Marine and biomass burning influence |
| AC20 | 01.10. | 7 h 05 min | 14.4 | Convection | Cloud profiling |

2.  **In section 4.3, the authors state that the sources of the organic aerosol in the LT and the UT are not the same, providing air mass trajectories along the flight track would help support these conclusions.**

We agree that air mass trajectories along the flight track help to support our conclusions. We analyzed FLEXPART trajectories, calculated along the flight track starting every minute and calculated backwards for 10 days providing hourly information on the location of each trajectory. Fig. S4 shows the release altitude of the trajectories against the residence time. The residence time gives the time that the trajectories spend in the boundary layer (BL, red curve) or in the upper troposphere (UT, black curve). There is only little interaction and almost no overlap between both curves. This leads to the conclusion that convection cannot be resolved with the FLEXPART model.

Fig. S5 shows maps with trajectories that are released below 4 km (lower troposphere, LT, Panel **(a)**) and above 8 km (upper troposphere, UT, Panel **(b)**). The origins of the trajectories differ overall. The trajectories released below 4 km have their origin also in the LT and show almost no interaction with higher air masses. Most of the trajectories come from the Atlantic Ocean and the southern part of South America. In contrast to this, the trajectories released above 8 km have their origin mainly above the Pacific Ocean and circulate at high altitudes above South America. Just a minor part of the trajectories origins from the eastern direction, coming from the Atlantic Ocean and/or Africa. Interactions with air masses at lower altitudes are rare, most prominent is the lifting at the Andes mountains.

We included the following to **Sect. 4.3, p. 15, line 12** (p. 16, line 12):
Air mass trajectories were calculated using the FLEXPART model. The trajectories are calculated along the flight tracks starting every minute and calculated backwards for 10 days providing hourly information on the location of each trajectory. The FLEXPART model is not able to resolve convective transport (see Fig. S4) for the ACRIDICON-CHUVA campaign. Nevertheless, the origin of the trajectories that are released in the LT (< 4 km) differs from the origin of the trajectories released in the UT (> 8 km) (see Fig. S5). The trajectories released in the LT have their origin also in the LT and show almost no interaction with higher air masses. Most of the trajectories come from the Atlantic Ocean and the southern part of South America. In contrast to this, the trajectories released above 8 km have their origin mainly above the Pacific Ocean and circulate at high altitudes above South America. Just a minor part of the trajectories origins from the eastern direction, coming from the Atlantic Ocean and/or Africa. Interactions with air masses at lower altitudes are rare, most prominent is the lifting at the Andes mountains.

We included the following to the Supplement:

[Figure]

Figure S4. Release altitude of the trajectories versus the residence time, i.e. the time that the trajectories spend in the boundary layer (BL, red curve) and in the upper troposphere (UT, black curve). There is only little interaction and almost no overlap between both curves. This leads to the conclusion that convection cannot be resolved with the FLEXPART model.

[Figure]

[Figure]

Figure S5. Maps with trajectories that are released below 4 km (lower troposphere, LT, Panel **(a)**) and above 8 km (upper troposphere, UT, Panel **(b)**). The colour code refers to the altitude of the centre trajectories.

3. **The authors mention that there are 4 CPC instruments operating during these flights, with cut of diameters of 4 and 10 nm. Were two of the CPCs at 4 nm and the other two at 10 nm. In the manuscript Andreae et al. (2018) it was mentioned that the second set of CPC instruments were coupled with a DMA set up for particle size distribution measurements. Is this also the case for these flights. According to the accompanying papers, other aerosol physical parameters should be available from the UHSAS for larger diameters. These size distribution measurements may help to support the conclusions on the SOA particle growth (Page 15, Line 13).**

We agree that these data would help to support our conclusions. The data were not finally processed during the manuscript preparation and therefore not used in the ACPD version of the manuscript. In the meantime, the validity of the UHSAS-A data was improved, so the UHSAS-A data are included now in the revised manuscript as follows:

**Sect. 2.1.2, p. 5, line 30** (p. 5, line 31):
For particles in the size range between 90 and 600 nm data from an ultra-high sensitivity aerosol spectrometer (UHSAS-A) that was installed as an underwing probe were analyzed. The measurement system is based on the detection of scattered light from laser illuminated aerosol particles. For the ACRIDICON-CHUVA flights used here (AC07-AC10 and AC15-AC20), the mentioned size range was divided into 66 logarithmic size bins. Data for the other four flights (AC11-AC14) are recorded in a different size binning and not used here. Cloud passages and intervals with sample flow deviations were removed. The UHSAS-A was calibrated using spherical polysterene latex particles.

**Sect. 4.3, p. 15, line 15** (p. 17, line 3):
Another indication supporting this is the size information of aerosol particles with diameters between 90 and 600 nm. Figure 6 shows the vertical profile of the median and the mode of the binned size distributions measured with the UHSAS-A (Panel **(a)**). It should be noted here, that the lowest cutoff of the considered size range of the UHSAS-A is at 90 nm. Accordingly, the displayed mode diameters are confined by this lower limit. Also, the displayed size distribution medians are affected by the size range limits and should only be interpreted in this context. Whereas in the LT the median and the mode are at diameters around 150 nm (median) and 130 nm (mode), respectively, both the median and the mode are shifted towards smaller diameters with increasing altitude. The lowest value of the median is reached at altitudes above 4 km and (apparently) remains constant. The colour code in the vertical profile refers to the size distributions for the three different altitude regions in Panel **(b)** of Fig 6. Shown are the median and interquartile range of the size distributions. The size distribution in the LT shows a maximum at 130 nm. The size distributions in the MT and UT are shifted towards smaller diameters, and it is clearly seen that the highest concentrations of small particles (around 90 nm) are found in the UT.

[Figure]

Figure 6. Vertical profile of the median (black dots) and the mode (triangles) of the binned size distributions (Panel **(a)**), and median and interquartile size distributions of particles between 90 and 600 nm in the UT (pink), in the MT (yellow) and in the LT (blue) (Panel **(b)**). The grey area in Panel **(a)** gives the interquartile range. The dotted line in Panel **(a)** indicates the lower cutoff of the considered size range of the UHSAS-A. The statistics shown in both Panels **(a)** and **(b)** are calculated from all valid UHSAS-A data from 10 flights (AC07-AC10, AC15-AC20). Data are calculated for STP conditions.

4. **Page 14, Line 6: It should be mentioned here that there is significant variability of the m/z 44 (or f44) among different AMS instruments and care should be taken when comparing results from different instruments (Frohlich et al., 2015, Pieber et al., 2016, Crenn et al., 2016).**

We agree to that and included the following to **Sect. 4.3, p. 14, line 13** (p. 15, line 25):

It should be mentioned here that there is a significant variability of m/z 44 (and $f_{44}$) among different AMS instruments such that no quantitative comparison can be done among the different data sets shown in Fig. 4 (Fröhlich et al., 2015, Crenn et al., 2015, Pieber et al., 2016).

5. **The authors detail several different methods to provide a robust characterization of the presence of organic nitrate and IEPOX SOA during these flights. It could be stated why PMF analysis was not used to try identify the presence of these aerosols. If IEPOXSOA is contributing up to 40% of the organic mass, they should be easily extracted by a PMF analysis. In addition, would adding inorganic ions (SO4 and NO3) into the PMF matrix help in extracting an organic nitrate factor that could then be compared with the other methods used to identify these species?**

In general, PMF can be used to deconvolute the organic matrix into several factors. However, in this study data from the entire campaign were analysed and cover a wide spatial area (horizontally

and vertically) with a fairly low temporal resolution due to high aircraft speed and low time resolution of the C-ToF-AMS. This is not the typical application of the PMF method which works best if a constant time series at constant location is analyzed. During our analysis we came to the conclusion that PMF analysis is not suited for the analysis of the data set presented here.

6. **For the calculation of the organic nitrate concentrations. It is not clear the difference the first estimation and the third estimation. These both methods are based on the ratio of the NO+/NO2+ ions in the instrument and how it varies from calibration values. However, using the method outlined in Kiendler-Scharr et al., is a more robust and tested method than just using the ratios alone. Can the authors comment on the added values of the first estimation compared to the third?**

The first estimation with the ratios alone can be seen as a demonstration that the potential for organic nitrates is given as it is the simplest method to check this. To give this study a more robust character the other methods were applied, compared, and discussed. According to a short comment (https://doi.org/10.5194/acp-2018-232-SC1) on our manuscript, a correction of the calculation was recommended and is included now.

7. **Clarification on the contents of Figures:**

    **Figure 2: presents data from all flights during the ACRIDICON-CHUVA campaign.**
    **Figure 3: presents data from 13 flights of the ACRIDICON-CHUVA campaign.**
    **Figure 4 to Figure 10: It is not stated which flights these measurements correspond.**
    **Can the authors provide more information on which flights were represented in figures 3 to 10 and why they were chosen over all 20 flights?**

The campaign consists in total of 14 flights. Fig. 2 shows the meteorological parameters during the campaign for all flights. 13 out of 14 flights were shown in Fig. 3 - 10. During one flight (AC10, conducted on 12.09.2014) no aerosol measurements of the background air were carried out. Therefore, data from this flight are missing for the AMS and are not shown in Fig. 3 – 12.
We included the following to **Sect. 4, p. 11, line 6** (p. 12, line 13):

One flight does not provide any aerosol data (AC10, conducted on 12.09.2014). Therefore, this flight is not included in the analysis of the C-ToF-AMS data. All figures are valid for 13 flights, except where otherwise noted.

**References**

Crenn, V., Sciare, J., Croteau, P. L., Verlhac, S., Fröhlich, R., Belis, C. A., Aas, W., Äijälä, M., Alastuey, A., Artiñano, B., Baisnée, D., Bonnaire, N., Bressi, M., Canagaratna, M., Canonaco, F., Carbone, C., Cavalli, F., Coz, E., Cubison, M. J., Esser-Gietl, J. K., Green, D. C., Gros, V., Heikkinen, L., Herrmann, H., Lunder, C., Minguillón, M. C., Močnik, G., O'Dowd, C. D., Ovadnevaite, J., Petit, J.-E., Petralia, E., Poulain, L., Priestman, M., Riffault, V., Ripoll, A., Sarda-Estève, R., Slowik, J. G., Setyan, A., Wiedensohler, A., Baltensperger, U., Prévôt, A. S. H., Jayne, J. T., and Favez, O.: ACTRIS ACSM intercomparison – Part 1: Reproducibility of concentration and fragment results from 13 individual Quadrupole Aerosol Chemical Speciation Monitors (Q-ACSM) and consistency with co-located instruments, Atmos. Meas. Tech., 8, 5063-5087, https://doi.org/10.5194/amt-8-5063-2015, 2015.

Fröhlich, R., Crenn, V., Setyan, A., Belis, C. A., Canonaco, F., Favez, O., Riffault, V., Slowik, J. G., Aas, W., Aijälä, M., Alastuey, A., Artiñano, B., Bonnaire, N., Bozzetti, C., Bressi, M., Carbone, C., Coz, E., Croteau, P. L., Cubison, M. J., Esser-Gietl, J. K., Green, D. C., Gros, V., Heikkinen, L., Herrmann, H., Jayne, J. T., Lunder, C. R., Minguillón, M. C., Močnik, G., O'Dowd, C. D., Ovadnevaite, J., Petralia, E., Poulain, L., Priestman, M., Ripoll, A., Sarda-Estève, R., Wiedensohler, A., Baltensperger, U., Sciare, J., and Prévôt, A. S. H.: ACTRIS ACSM intercomparison – Part 2: Intercomparison of ME-2 organic source apportionment results from 15 individual, co-located aerosol mass spectrometers, Atmos. Meas. Tech., 8, 2555-2576, https://doi.org/10.5194/amt-8-2555-2015, 2015.

Simone M. Pieber, Imad El Haddad, Jay G. Slowik, Manjula R. Canagaratna, John T. Jayne, Stephen M. Platt, Carlo Bozzetti, Kaspar R. Daellenbach, Roman Fröhlich, Athanasia Vlachou, Felix Klein, Josef Dommen, Branka Miljevic, José L. Jiménez, Douglas R. Worsnop, Urs Baltensperger, and André S. H. Prévôt (2016) Inorganic salt interference on CO2+ in aerodyne AMS and ACSM organic aerosol composition studies. Environmental Science & Technology 2016, 50 (19), 10494-10503. DOI: 10.1021/acs.est.6b01035.

---

## Author Comment (AC3) · 14 Aug 2018

We thank Referee #2 for her/his comments and suggestions that helped to improve the manuscript. Our response is formatted as follows:

**Reviewer's comments**

Author's reply

Changes to the manuscript

All page, line, section and figure numbers in bold refer to the original manuscript, all others to the revised version.

**This paper reported an aircraft measurement in the tropical troposphere over the amazon region. The vertical profile of the main components in submicron aerosols were measured by a C-ToF-AMS. The authors focus their discussion on SOA formed through isoprene oxidation under low NO condition, i.e. isoprene epoxydiols-derived SOA (IEPOX-SOA) and organic nitrate formation. Vertical profiles of IEPOX-SOA and organic nitrate mass up to 14 km are new and very interesting. However, the analysis method of this paper has a very serious caveat which needs to be furtherly explored and addressed. I recommend a major revision due to the comments as below:**

1. **The entire analysis of this paper only depends on AMS measurement. To prove the deduction that the IEPOX-SOA can be formed in the upper troposphere. The authors should show evidences. E.g., what are the vertical profile of isoprene and NO, or even gas-phase IEPOX. The authors argued there is high organic nitrate formation due to NOx production of lighting. However, the high NO will inhibit IEPOX-SOA formation (Paulot, Crounse et al. 2009, de Sá, Palm et al. 2016, Liu, Brito et al. 2016). The addressed reason for organic nitrate formation conflicts with that for IEPOX-SOA formation. And the authors should show recalculate the PH after correcting the mass concentration of nitrate. See detailed information in the followed comment.**

   Isoprene or gas-phase IEPOX were not measured during the ACRIDICON-CHUVA campaign as there were no appropriate instruments on board. The vertical profile of NO and $NO_y$ can be found in **Fig. 8** (Fig. 9, p. 21). We also included UHSAS-A data to provide size information of particles with a diameter between 90 and 600 nm.
   We agree to the statement that high NO will inhibit the formation of IEPOX-SOA as already mentioned in our manuscript on **p. 16, line 24**. We concluded that the oxidation of isoprene to IEPOX must occur in regions with low NO, e.g. during the vertical transport to high altitudes.
   A recalculation for the acidity of aerosol particles after nitrate correction was done and Fig. S3 in the supplement is updated.

2. **This paper reported the mass and ions measured with C-ToF-AMS, which gives an UMR spectrum. M/z 30 is mainly composed of CH2O+ and NO+ ions, and m/z 46 is NO2 and CH2O2+. The contribution of CH2O ions to m/z 30 (CH2O2 to m/z 46) is very high in areas strongly influenced by biogenic emissions. In the SE US, the aerosol composition of which is very similar to the Amazon area, UMR nitrate (with NO+, NO2, CH2O+CH2O2) is overestimated a factor of 2-3 than the high-resolution nitrate (true nitrate with NO+ and NO2) based on the default fragmentation table, due to underestimation of the organic (CH2O and CH2O2) interferences (Hu, Campuzano-Jost et al. 2017). Thus the nitrate mass concentration reported in this study are**

**combination of organic and nitrates. As a result, the ratio of UMR m/z 30 and UMR 46 cannot be used to calculate organic nitrate mass concentration, unless the authors find a way to exclude the interference. A revised fragmentation table is suggested in (Allan, Morgan et al. 2014). However, its suitability for Amazon area still needs to be evaluated. Similarly, the authors cannot really report the IEPOX-SOA increases with nitrate, which is probably organics. The statement in page 17 1-7 lines are wrong due to the reason mentioned here.**

As already suggested by the short comment (https://doi.org/10.5194/acp-2018-232-SC1) on our manuscript, a correction of the calculation of organic nitrate was recommended. This was done according to a recent publication by Fry et al., 2018 (https://doi.org/10.5194/acp-2018-255) and is included now. We added the following to the manuscript to **Sect. 3.4 p. 9, line 16** (p. 10, line 9):

As organic interferences on the mass spectral signals at m/z 30 (interference from $CH_2O^+$) and m/z 46 (interference from $CH_2O_2^+$) can occur in environments with high biogenic contribution and/or small nitrate concentrations, a correction according to Fry et al. (2018) was applied. The correction of both signals at m/z 30 and 46 is achieved by using correlated organic signals at m/z 29, 42, 43, and/or 45 derived by high resolution measurements. The organic signals at m/z 29 ($CHO^+$) and m/z 45 ($CHO_2^+$) are closest to those affected by the interference and used for the correction here. Equations 4 and 5 give the individual correction for the nitrate signal at m/z 30 and 46, respectively. The correction for $NO^+$ includes the total signal at m/z 30, the default fragmentation correction from the air signal (Allan et al., 2004), and a correction coefficient that depends on the m/z used for the correction ($A_i$). As for m/z 30 the correlated organic signal at m/z 29 is used here, the organic signal at m/z 29 (Org29) needs to be taken into account as well as the contribution of the isotopes of organic CO. For the correction of the nitrate fraction at m/z 46 a term which includes a correlation coefficient $B_i$ and the organic signal at m/z 45 is subtracted from the signal at m/z 46. The correction coefficient $A_i$ is in this case 0.215, $B_i$ is 0.127 (see supplement to Fry et al., 2018). In the organic signal at m/z 28, 29, 30 and 45 the relative ionization efficiency ($RIE_{Org}$) is already applied and needs to be reversed for the correction of the nitrate signal.

Nitrate fraction at m/z 30:
$NO^+ = m/z\ 30 - 0.0000136 * m/z\ 28 - A_i * (Org29 - 0.011 * Org28) * RIE_{Org} - Org30 * RIE_{Org}$ (4)

Nitrate fraction at m/z 46:
$NO_2^+ = m/z\ 46 - B_i * Org45 * RIE_{Org}$ (5)

The total nitrate signal is then calculated by adding both fractions. The final nitrate mass concentrations were reduced by 0.045 µg m$^{-3}$ (STP) on average corresponding to an averaged reduction of 39 % of the initial nitrate mass concentrations. A comparison of the initial and finalized nitrate mass concentrations can be found in the Supplement (see Fig. S2).

Furthermore, we added the following to the paragraph in **Sect. 4.5 p. 19, line 2** (p. 21, line 16):

Figure 10 shows the scatter plot of the corrected $NO^+$ and $NO_2^+$ for the LT (dark blue) and the UT (light blue). The linear fit curves for the LT and UT have an intercept of 0, proving that the applied correction is essential. Nevertheless, it has to be noted that the correction is based on correlations between different m/z signals derived from measurements at low altitudes (Fry et al., 2018), thus the application to UT data bears uncertainties, because the conditions (especially temperature)

are different. However, high resolution AMS measurements at these altitudes are currently not available.

The linear fit for the LT data shows a higher slope than that derived from calibrations with ammonium nitrate. The linear fit for the UT shows also a higher slope and some data points are still significantly above the fitted ratio between $NO^+$ and $NO_2^+$. This can be seen as a first hint that organic nitrates might be observed, especially in the UT.

[Figure]

Figure 10. Scatter plot of $NO^+$ and $NO_2^+$ for the lower troposphere (LT, dark blue) and the upper troposphere (UT, light blue). The nitrate signals have been corrected for organic interference according to Fry et al., 2018. Linear fit curves are shown in red, the ratio of $NO^+$ and $NO_2^+$ derived from calibrations with ammonium nitrate is presented by the red dashed line.

We added the following to the supplement:

[Figure]

Figure S2. Vertical profile of median nitrate mass concentrations and interquartile ranges before (light blue) and after (dark blue) corrections according to Fry et al., 2018 and supplement to Fry et al., 2018.

Furthermore, we changed the paragraph in **Sect. 4.4 p. 17, line 2** (p. 20, line 8) to the following:

Although the correlation between IEPOX-SOA and nitrate is weak, it may indicate that not only sulfate, but also nitrate can provide the acidic conditions for the partitioning of IEPOX to IEPOX-SOA. For our data, taking only the inorganic species (nitrate and sulfate) into account for acidity calculations, the aerosol is mainly neutralized (see Fig. S6). Although there is a tendency in the UT above 10 km that the measured ammonium is not sufficient to neutralize the inorganic species, a quantitative statement cannot be made as the values fall below or close by the DL. The presence of organosulfates and -nitrates could also affect the acidity calculations. As for the organonitrates the data are already corrected, for organosulfates such a similar correction is not possible with

data from a C-ToF-AMS as there are no different fragmentation patterns between inorganic and organic sulfates (Farmer et al., 2010).

3. **Page 20 line 15, there is no way that the calculated organic nitrate mass concentration based on Fry et al. (2013) can be higher than the total nitrate mass concentration measured from AMS. Because Fry et al. (2013) used a method base on splitting the mass of total nitrate masses with differential NO/NO2 ratios from NH4NO3 and organic nitrate. Please check furtherly.**

After correcting the contribution of organics to m/z 30 and m/z 46 all data were recalculated. For the comparison with the method described by Fry et al. (2013) a good agreement is now achieved. The vertical profiles show similar median values (see Fig. 11). Differences in the scatter plot are in the range of uncertainty of the data (see Fig. 12).

4. **How are the vertical profiles of main components in submicron aerosols in this study compared to the amazon study reported in (Allan, Morgan et al. 2014). Also the It will be interesting if the authors can address the similarity and differences, especially for m/z 82 and f82. What is the new finding in this study compared to the previous ones.**

The study by Allan et al., 2014 reported aircraft measurements above the Amazon rain forest at altitudes up to 5 km. Our study presents also data from higher altitudes up to 14 km, providing more information on the vertical profile of submicron aerosol chemical composition and $f_{82}$.
We added a short comparison with the study by Allan et al., 2014 to the paragraph in **Sect. 4.4 p. 16, line 10** (p. 18, line 12):

To the best of our knowledge, $f_{82}$ data from the tropical upper troposphere have not yet been reported in the literature. Aircraft measurements above the Amazon rainforest were reported by Allan et al. (2014), but these data are restricted to altitudes below 5 km, corresponding to the definition of the LT in this study. In Allan et al. (2014), two cases are presented and show background (4 ‰) and increased values (9 ‰) of $f_{82}$. The highest values of $f_{82}$ were found on top of the boundary layer, decreasing with increasing altitudes. These values are similar to the data from the LT presented here.

**Minor comments:**

5. **Page 7 line 12: please give details about the "a time-dependent cubic spline function was used to determine a detection limit for each data point"**

In order to provide more details on the function that was used to determine the detection limits, we added the following to the manuscript to **Sect. 3.1 p. 7, line 11** (p. 7, line 13).

Here, a time-dependent cubic spline function was used to determine a detection limit for each data point. This function was developed at the Max Planck Institute for Chemistry (MPIC) and is introduced and explained in more detail in a PhD thesis (Reitz, 2011). For every data point of the background signal a third order polynomial (cubic) function is calculated through the four neighbouring points (two before and two behind) while omitting the actual point. Applying this method, all trends from the background signal are excluded and just the short-term noise remains.

A quantity $R$ is introduced in the algorithm and characterizes the statistical spread of the noise. $R$ is defined by the squares of the deviation between the omitted centre point and the cubic function along a moving window. To relate this $R$ to the standard deviation ($\sigma$) a proportionality factor of $(18/35)^{1/2}$ is needed (Reitz, 2011). The exact derivation of the proportionality factor will not be explained here, but qualitatively it accounts for the fact that not only the points are affected by noise but also the cubic function itself. Thus, $R$ is larger than the standard deviation ($\sigma$). This calculation also provides a continuous $\sigma$ from which the detection limit can be derived using the equation DL = 3 x √2 x $\sigma$.

6. **Page 6 line 13: give full citation of Molleker et al. 2018.**

   Molleker et al., 2018 is a manuscript in preparation. We changed the citation in the manuscript to the following:

   Further details of the constant pressure inlet are subject of a separate publication (Molleker et al., manuscript in preparation).

7. **Page 8 line 15-16: Hu et al. 2015 showed an f82 value from IEPOX-SOA and background sources in the appendix.**

   We do not understand this comment. We refer to Hu et al. (2015) already.

8. **Page 8 equation 1, what is the f82 values of IEPOX-SOA used in this study to calculate the mass concentration of IEPOX-SOA.**

   The $f_{82}$ value of IEPOX-SOA used in our study is 22 ‰ as suggested in Hu et al. 2015. We denoted this as $f_{82}^{IEPOX-SOA}$ (**p. 8, line 26-27** (p. 9, line 7-8)).

**References:**

Allan, J. D., Morgan, W. T., Darbyshire, E., Flynn, M. J., Williams, P. I., Oram, D. E., Artaxo, P., Brito, J., Lee, J. D., and Coe, H.: Airborne observations of IEPOX-derived isoprene SOA in the Amazon during SAMBBA, Atmos. Chem. Phys., 14, 11393-11407, https://doi.org/10.5194/acp-14-11393-2014, 2014.

Fry, J. L., Brown, S. S., Middlebrook, A. M., Edwards, P. M., Campuzano-Jost, P., Day, D. A., Jimenez, J. L., Allen, H. M., Ryerson, T. B., Pollack, I., Graus, M., Warneke, C., de Gouw, J. A., Brock, C. A., Gilman, J., Lerner, B. M., Dubé, W. P., Liao, J., and Welti, A.: Secondary Organic Aerosol (SOA) yields from $NO_3$ radical + isoprene based on nighttime aircraft power plant plume transects, Atmos. Chem. Phys. Discuss., https://doi.org/10.5194/acp-2018-255, in review, 2018.

Hu, W. W., Campuzano-Jost, P., Palm, B. B., Day, D. A., Ortega, A. M., Hayes, P. L., Krechmer, J. E., Chen, Q., Kuwata, M., Liu, Y. J., de Sá, S. S., McKinney, K., Martin, S. T., Hu, M., Budisulistiorini, S. H., Riva, M., Surratt, J. D., St. Clair, J. M., Isaacman-Van Wertz, G., Yee, L. D., Goldstein, A. H., Carbone, S., Brito, J., Artaxo, P., de Gouw, J. A., Koss, A., Wisthaler, A., Mikoviny, T., Karl, T., Kaser, L., Jud, W., Hansel, A., Docherty, K. S., Alexander, M. L., Robinson, N. H., Coe, H., Allan, J. D., Canagaratna, M. R., Paulot, F., and Jimenez, J. L.: Characterization of a real-time tracer for isoprene epoxydiols-derived secondary organic

aerosol (IEPOX-SOA) from aerosol mass spectrometer measurements, Atmos. Chem. Phys., 15, 11807-11833, https://doi.org/10.5194/acp-15-11807-2015, 2015.

Reitz, P. (2011). Chemical composition measurements of cloud condensation nuclei and ice nuclei by aerosol mass spectrometry. Mainz: University.
url: http://nbn-resolving.org/urn:nbn:de:hebis:77-28691.